



# Increasing the spatial resolution of cloud property retrievals from Meteosat SEVIRI by use of its high-resolution visible channel: implementation and examples

Hartwig Deneke[1], Carola Barrientos-Velasco[1], Sebastian Bley[2], Anja Hünerbein[1], Stephan Lenk[1], Andreas Macke[1], Jan Fokke Meirink[3], Marion Schroedter-Homscheidt[4], Fabian Senf[1], Ping Wang[3], Frank Werner[5], and Jonas Witthuhn[1]

[1]Leibniz Institute for Tropospheric Research, Permoserstraße 15, 04318 Leipzig, Germany
[2]ESA Centre for Earth Observation, Largo Galileo Galilei, 1, 00044 Frascati RM, Italy
[3]Royal Netherlands Meteorological Institute, Utrechtseweg 297, 3731 GA De Bilt
[4]German Aerospace Center (DLR), Institute of Networked Energy Systems, Carl-von-Ossietzky-Straße 15, 26129 Oldenburg, Germany
[5]Jet Propulsion Laboratory, 4800 Oak Grove Drive, Pasadena, CA 91109, USA

**Correspondence:** Hartwig Deneke, deneke@tropos.de

**Abstract.** The modification of an existing cloud property retrieval scheme for the Spinning Enhanced Visible and Infrared Imager (SEVIRI) instrument onboard the geostationary METEOSAT satellites is described to utilize its high-resolution visible (HRV) channel for increasing the spatial resolution of its physical outputs. This results in products with a nadir spatial resolution of $1 \times 1 \, \text{km}^2$, compared to the standard $3 \times 3 \, \text{km}^2$ resolution offered by the narrowband channels. This improvement thus

5   greatly reduces the resolution gap between current geostationary and polar-orbiting meteorological satellite imagers. In the first processing step, cloudiness is determined from the HRV observations by a threshold-based cloud masking algorithm. Subsequently, a linear model that links the $0.6 \, \mu\text{m}$, $0.8 \, \mu\text{m}$, and HRV reflectances provides a physical constraint to incorporate the spatial high-frequency component of the HRV observations into the retrieval of cloud optical depth. The implementation of the method is described, including the ancillary datasets used. It is demonstrated that the omission of high-frequency variations

10  in the cloud-absorbing $1.6 \, \mu\text{m}$ channel results in comparatively large uncertainties in the retrieved cloud effective radius, likely due to the mismatch in channel resolutions. A newly developed downscaling scheme for the $1.6 \, \mu\text{m}$ reflectance is therefore applied to mitigate the effects of this scale mismatch. Benefits of the increased spatial resolution of the resulting SEVIRI products are demonstrated for three example applications: (i) for a convective cloud field, it is shown that significantly better agreement between the distributions of cloud optical depth retrieved from SEVIRI and from collocated MODIS observations is

15  achieved; (ii) the temporal evolution of cloud properties for a growing convective storm at standard and HRV spatial resolutions are compared, illustrating an improved contrast in growth signatures resulting from the use of the HRV channel; (iii) an example of surface solar irradiance, determined from the retrieved cloud properties, is shown, where the HRV channel helps to better capture the large spatio-temporal variability induced by convective clouds. These results suggest that incorporating the HRV channel in the retrieval has potential for improving METEOSAT-based cloud products for several application domains.



## 1  Introduction

Clouds play an important role in Earth's energy budget and hydrological cycle (e.g, Wild and Liepert, 2010). Despite their importance, the representation of clouds in current climate and weather forecast models remains limited due to a fundamental

lack of understanding of the relevant cloud processes and the interaction of clouds with other components of the climate system (Bony et al., 2015). These shortcomings are widely recognized to be a dominant source of uncertainty in our understanding of the climate system, and its response to anthropogenic forcings (Boucher et al., 2013).

Due to their excellent spatial and temporal coverage, as well as the multi-decadal length of observational data records, multi-spectral meteorological satellite imagers offer a unique source of information for studying the role of clouds in the

climate system. The underlying methods for inferring cloud properties from these instruments are, however, usually non-linear and sensitive to assumptions and uncertainties in the applied forward models, which reflects the underconstrained nature of the underlying inversion problem (Stephens and Kummerow, 2007). This introduces sensitivities of the resulting products to sensor characteristics, such as spectral response and spatial resolution, as well as relatively minor differences in the implementation of retrieval algorithms (Roebeling et al., 2015). As a result, inhomogeneities and inconsistencies can be found in cloud data

records from different satellite platforms or data providers (e.g., Karlsson and Devasthale, 2018).

Currently, the Moderate Resolution Imaging Spectroradiometer (MODIS) flown on the polar-orbiting satellites Terra and Aqua are one of the most widely used satellite instruments for studying the role of clouds in the climate [1]. Based on the method described by Nakajima and King (1990), cloud products (e.g., estimates of cloud phase, optical depth, effective radius, and cloud water path) are provided at a spatial resolution of $\sim 1 \times 1\,\mathrm{km}^2$ (Platnick et al., 2003). Despite their wide use, it is

well recognized that sub-pixel variability and 3D radiative effects can introduce substantial biases and uncertainties in these products, which depend on various factors such as solar and viewing geometry (see e.g., Cahalan et al., 1994; Marshak et al., 2006; Zhang et al., 2012; Horváth et al., 2014). While the retrieval of cloud droplet number concentration is of high scientific interest due to its relevance for elucidating the climate impact of aerosol-cloud interactions, it is particularly challenging (Grosvenor et al., 2018).

In contrast to polar-orbiting satellites, observations from a geostationary orbit allow to fully resolve the diurnal cycle (e.g., Seethala et al., 2018), and to study the temporal evolution of shallow and deep convective clouds (e.g., Bley et al., 2016; Senf et al., 2015). They are also widely used to derive surface solar irradiance (SSI) as needed by the solar energy community (e.g., Tarpley, 1979; Möser and Raschke, 1984; Cano et al., 1986; Rigollier et al., 2004; Greuell et al., 2013; Qu et al., 2017).

While the recently launched geostationary Himawari and GOES-R series satellites carry instruments with a similar spectral

response and spatial resolution to MODIS for their solar channels (Miller et al., 2016; Schmit et al., 2017), such observations are not yet available over Europe, where the third generation of METEOSAT with similar spatial resolution capabilities is

---

[1]A search of Google Scholar ™, https://scholar.google.com for the term cloud combined with MODIS yields 167.000 hits, compared to 58.300 hits for the term AVHRR, 10.800 for SEVIRI, and 9.930 for VIIRS (Date of access: 3 August 2020)





scheduled for launch in 2022. The current operational second generation METEOSAT satellites are equipped with the Spinning Enhanced Visible and Infrared Imager (SEVIRI) instrument, with an at the present time comparatively coarse nadir spatial resolution of $3 \times 3\,\mathrm{km}^2$ for its narrowband spectral channels (Schmetz et al., 2002). Considering the start of SEVIRI's operational service in 2004, and thus the length of its observational record, as well as its relatively advanced sensor characteristics,

SEVIRI data remains of high interest both for scientific investigations in atmospheric and climate sciences, and for usage by solar energy industries.

In addition to its 11 narrow-band channels, the SEVIRI instrument features a high resolution visible (HRV) channel with a nadir resolution of $1 \times 1\,\mathrm{km}^2$. The HRV channel is widely used to derive SSI based on cloud index-based methods (e.g., Rigollier et al. (2004) as used in Helioclim-3 [2]; Hammer et al. (2003) as used in EnMetSol [3]; or Pfeifroth et al. (2019)). Despite

some previous studies (e.g., Klüser et al., 2008; Carbajal Henken et al., 2011), a systematic and quantitative use of the HRV channel for producing cloud property datasets based on physical retrievals has not been pursued. The present article aims to fill this gap, by extending and combining the methods introduced in Deneke and Roebeling (2010) and Bley and Deneke (2013), and introducing techniques for utilizing the HRV channel together with SEVIRI's other channels for quantitative cloud retrievals. This effort builds on the well-established Cloud Physical Property (CPP) retrieval described by Roebeling et al.

(2006), which is also utilized at the heart of the CLAAS-1 and CLAAS-2 climate data records (Stengel et al., 2014; Benas et al., 2017) provided by the Climate Monitoring Satellite Application Facility (CM SAF, Schulz et al., 2009).

By applying the Surface Irradiance for Cloudy Conditions from SEVIRI (SICCS) algorithm (Deneke et al., 2008; Greuell et al., 2013) to the improved cloud properties at HRV resolution, corresponding solar irradiances at the surface and top-of-atmosphere (TOA) can also be retrieved with this scheme. The simultaneous and consistent retrieval of cloud properties

and irradiances can help to answer the question whether higher spatial resolution of satellite observations can improve the agreement with ground-based cloud and irradiance measurements. It may also help to quantify which accuracy can be achieved for different types of clouds, and which physical mechanisms are responsible for deviations.

The present paper is a companion paper to Werner and Deneke (2020), which focuses on the methodological choices and details of the downscaling algorithm for the SEVIRI reflectances and presents an evaluation of the accuracy of the retrieved

cloud products using MODIS observations as reference. In contrast, the present paper gives an overview of the complete retrieval setup, including its ancillary inputs, and describes three applications which might potentially benefit from the enhanced resolution.

The paper is structured as follows: in Sec. 2, the relevant instrumental characteristics of Meteosat SEVIRI used as basis of this paper are introduced. Sec. 3 summarizes the various steps in the overall processing scheme. Special attention is given

to describe the modifications required to utilize the HRV channel. Sec. 4 presents some example applications and illustrates the benefits resulting from the increased spatial resolution. Sec. 5 closes by drawing conclusions and presenting an outlook to future work.

---

[2]See http://www.soda-pro.com/de/help/helioclim/helioclim-3-overview, last access: 3 August 2020

[3]See https://uol.de/en/energiemeteorology/services/enmetsol, last access: 3 August 2020





## 2 Instrumental Data

[Figure 1 about here.]

Meteosat Second Generation is the current series of European Geostationary Weather Satellites operated by the European Organisation for the Exploitation of Meteorological Satellites (EUMETSAT). Four MSG satellites have been launched since
2003 and carry SEVIRI as main payload (Schmetz et al., 2002). While the primary objective of MSG is to acquire full-disk imagery for meteorological applications at $0°$ longitude with a $15$ minute repeat cycle, the backup satellites provide the so-called Rapid Scan Service (RSS) since 2008, which covers Europe with an enhanced 5-minute repeat cycle at a nominal sub-satellite longitude of $9.5°$ East.

The SEVIRI instrument has three solar and eight infrared narrowband spectral channels with a nadir sampling resolution of
$3 \times 3\,\mathrm{km}^2$. In addition, the high resolution visible (HRV) channel offers an increased spatial resolution of $1 \times 1\,\mathrm{km}^2$ at nadir, but at the cost of a relatively broad spectral response resembling that of the first generation of Meteosat satellites (Cros et al., 2006). A further limitation is the fact that HRV images are only available for half of the nominal field of view of the narrowband channels due to its high data volume.

Fig. 1 shows the spectral response of the solar channels in panel (a), while the modulation transfer function (MTF) in North-
South and East-West direction (as provided by EUMETSAT, 2012) is displayed in panel (b). The vertical black lines correspond to the Nyquist frequencies for the sampling resolution of the respective channels. The MTF describes the attenuation of the amplitude of a sine-like pattern as a function of frequency and is linked to the spatial response through the Fourier transform (see Deneke and Roebeling, 2010, for details). The difference of the MTFs of the HRV and the $0.6\,\mu\mathrm{m}$ channel is also plotted in the figure. It is used in our method to extract the high spatial frequency component contained in the HRV channel observations
which is not resolved by the lower-resolution channels, as is described in detail in Sec. 3.5 of the paper. It should be noted that the optical resolution of the SEVIRI channels is lower than their sampling resolution by a factor of about 1.6, which can be seen by the significant attenuation of the frequency response well below the Nyquist limit. This causes oversampling and implies a significant increase of the effective area sampled by each pixel, compared to the area calculated from the sampling resolution (Schmetz et al., 2002). Moreover, for higher-latitude regions the spatial resolution is also reduced due to the oblique
viewing angle of Meteosat, resulting in an increase of the pixel extent by roughly a factor of two in North-South direction for observations over Germany.

EUMETSAT provides an operational calibration of SEVIRI images obtained by a vicarious calibration technique (Govaerts et al., 2004). Meirink et al. (2013) confirm the temporal stability of this calibration, but find relatively large systematic differences of up to $8\,\%$ for collocated near-nadir reflectances from the SEVIRI and MODIS instruments. Channel-specific correction
factors to account for these differences have been derived and are applied by the CM SAF for generating SEVIRI-based climate datasets (e.g. Benas et al., 2017), based on the expectation of the authors that the MODIS calibration is more reliable. Here, the decision has been made to adopt the same correction factors also for the retrievals within the scope of the present study.

The primary objective of the Meteosat satellites is to support short-range weather forecasting in general, and forecasting of rapidly developing high impact weather events in particular. In addition, its observations are currently used in a wide





range of applications in meteorology, hydrology, and climatology. Among these, the monitoring and nowcasting of surface solar irradiance for estimating photovoltaic or solar thermal power yield is a growing field of relevance for the transition to renewable energy.

## 3 Retrieval Scheme

5                                   [Figure 2 about here.]

This section presents a description of the improved processing scheme, which has been established for retrieving cloud products and solar irradiance at the TOA and surface from Meteosat SEVIRI at HRV resolution.

An overview of the overall workflow is given in Fig. 2. This scheme builds upon software packages developed within the framework of the Satellite Application Facility on Support to Nowcasting and Very Short Range Forecasting (NWC SAF, Fernandez et al., 1999) and on Climate Monitoring (CM SAF, Schulz et al., 2009). While the former software is utilized in unmodified form, the latter CPP code (Roebeling et al., 2006) has been adapted to work on input data interpolated to the grid of the HRV channel, and to exploit the additional information content on high-frequency spatial variability captured by the HRV channel, which is not resolved by the lower-resolution narrowband channels. For this purpose, the HRV channel reflectances are filtered by a high-pass filter, thereby removing the variability already resolved by the lower-resolution narrowband channels, and are then used as additional inputs. Also, the normal CPP input fields consisting of SEVIRI level 1.5 radiances, the NWCSAF cloud mask, type, and height products, as well as the ancillary data, are interpolated to the spatial grid of the HRV channel. Consistent with Deneke and Roebeling (2010), trigonometric interpolation is used for the radiances, while nearest-neighbor (N.N.) interpolation is used for the cloud products and bilinear interpolation for the ancillary datasets. In addition, a high-resolution cloud masking algorithm is applied to the HRV channel reflectances, which has been introduced previously by Bley and Deneke (2013). In the final step of the processing chain, these cloud products are used in combination with ancillary data on surface albedo, water vapor, and ozone column values by the Surface Insolation under Clear and Cloudy skies derived from SEVIRI imagery scheme (SICCS, Deneke et al., 2008; Greuell et al., 2013) to obtain estimates of the clear-sky and cloudy sky solar irradiances at TOA and the bottom-of-atmosphere.

[Figure 3 about here.]

25       In its current form, the retrieval scheme has been set up for a processing region of $240 \times 400$ standard-resolution pixels centered on Germany and Central Europe. The domain can be seen in Fig. 3. To complement the improved spatial resolution with the highest possible temporal resolution, Meteosat's Rapid Scan Service is used as the primary input data stream. The relatively small processing region has been chosen to keep the processing time within reasonable bounds, as the pixel number increases by a factor of 9. Combined with the increased complexity in the algorithms introduced by the use of the HRV channel, the overall processing time is larger by about a factor of 15 compared to an identical low-resolution region.

More detailed descriptions of the individual steps of the processing and the used ancillary datasets are given in the following subsections.



### 3.1 NWCSAF Cloud Products

Within the improved scheme, several cloud products are generated from MSG SEVIRI observations based on the NWC SAF software in its 2016 version. From the NWC SAF cloud product suite, the cloud mask (CMa) and cloud type (CT), as well as cloud top pressure (CTP), height (CTH), and temperature (CTT) (Derrien and Le Gléau, 2005) are used. The cloud masking

and typing schemes apply a set of multi-spectral threshold tests to derive an objective classification of the observed satellite scenery. Thresholds depend on solar illumination, viewing geometry and a reasonable guess of the atmospheric state. For this purpose, the software provides climatological average values, but can also employ forecast or reanalysis fields from numerical weather prediction models for an improved accuracy. The cloud mask provides a binary distinction between cloud-free and cloud-contaminated or -filled pixels. Cloud typing divides cloud-containing satellite pixels into categories based on their height

and opacity. Opaque clouds are classified into five classes: very low, low, medium, high, and very high clouds, which are separated by the pressure levels of 800, 650, 450, and 300 hPa. Additional classes distinguish between fractional clouds and high cirrus clouds with varying levels of opacity. The NWCSAF software currently does not separate between convective and stratiform cloud structures. The cloud-top height algorithm uses the radiative transfer code RTTOV (Saunders et al., 2018) as forward model to relate the observed infrared radiances to those of a simulated cloud at a given vertical position and for given

atmospheric conditions to infer the height of the top of clouds. For semi-transparent clouds, either a CO2 slicing or a water vapor intercept method is applied (Borde et al., 2004).

### 3.2 Cloud Physical Properties Retrieval

Following the physical principles described by Nakajima and King (1990), the CPP algorithm uses a measured reflectance pair at a visible and a shortwave infrared (SWIR) wavelength, in this case from the SEVIRI $0.6\,\mu$m and $1.6\,\mu$m channels, to retrieve

the cloud optical thickness ($\tau$) and effective particle radius ($r_e$). Retrievals are performed for either a liquid or an ice cloud, based on a determination of the cloud phase by a modified version of the Pavolonis et al. (2005) algorithm, which is described in more detail in Benas et al. (2017).

CPP employs precalculated lookup tables (LUTs) of TOA cloud-top reflectances in a Rayleigh atmosphere, which have been simulated by the Doubling-Adding KNMI (DAK) radiative transfer model (Stammes, 2001). Details on the underlying

single-scattering properties of liquid and ice cloud particles can be found in Benas et al. (2017). The measured reflectances are corrected for absorption by atmospheric gases based on Moderate Resolution Atmospheric Transmission (MODTRAN4 version 3, Anderson et al., 2001) simulations. Subsequently, a match between the measurements and the LUT values of simulated reflectances is sought, which yields the cloud optical properties $\tau$ and $r_e$. Uncertainties of the retrieved values are estimated based on a $3\%$ relative error in the reflectances. A range of ancillary data is needed, including surface reflectance in

the SEVIRI channels, as well as vertically integrated column values of water vapor and ozone. The datasets used in the present scheme are listed in Section 3.4. Extensive details on the retrieval algorithm can be found in CMSAF (2016).





### 3.3 Solar Irradiance

The SICCS algorithm (Deneke et al., 2008; Greuell et al., 2013) estimates solar irradiances at the TOA and surface from LUTs, which are calculated using the DAK model for cloud-free conditions, as well as for water and ice clouds. Both direct and global irradiances are calculated at the surface. A broadband version of DAK has been used for the calculation of the underlying LUTs,

employing the correlated k-distribution technique to account for atmospheric gas absorption (Kuipers Munneke et al., 2008). For cloud-free pixels, aerosol properties (optical depth, Ångström parameter, and single scattering albedo) are considered as input parameters, together with surface elevation, to estimate the resulting clear-sky irradiances. For cloudy pixels, the atmospheric transmission is further adjusted to account for the effects of cloud phase, $\tau$, and $r_e$. For both cloudy and cloud-free pixels, the water vapor column, total ozone column, and surface albedo are taken into account and thus have to be provided

as inputs.

Greuell et al. (2013) performed an extensive validation of the surface irradiance retrievals with European Baseline Surface Radiation Network (BSRN) measurements. Across all European sites, a median bias of 7 W m$^{-2}$ (2%) and a root mean square error (RMSE) of 65 W m$^{-2}$ (18%) was found for hourly values of surface irradiance.

### 3.4 Ancillary Datasets

A number of ancillary input datasets have been modified in comparison to the original CPP/SICCS retrievals. The land-sea mask has been generated from the current version 2.3.7 of the Global Self-consistent Hierarchical High-resolution Shoreline (GSHHS) dataset (Wessel and Smith, 1996) for the considered region and the HRV grid. Surface elevation has been obtained from the SRTM15_PLUS digital elevation model (Tozer et al., 2019), which is based on the NASA Shuttle Radar Topography Mission (SRTM). It is available at a spatial resolution of 15 seconds (approximately 500 m), which is slightly higher than that

of the HRV grid. Using this dataset, the much larger data volume of the original SRTM digital elevation models available at 30 m and 90 m resolutions can be avoided.

Both the NWCSAF and CPP algorithms are configured to use Numerical Weather Prediction input fields to account for the effects of the current atmospheric temperature and humidity profiles on infrared radiances by use of the RTTOV radiative transfer model. As input data streams, either the CAMS (Copernicus Atmospheric Monitoring Service) reanalysis (Inness et al.,

2019) or the operational ECMWF forecast can be used alternatively with 3-hourly resolution, the latter allowing for near-real time processing, while the former is only available with some time delay. In addition, the NWCSAF software uses the OSTIA dataset as input for sea surface temperature (Donlon et al., 2012). Spectral and broadband surface reflectance maps from the Land Surface Analysis Satellite Application Facility (LSA SAF, Carrer et al., 2018) are used in our modified version of the CPP algorithm to account for the effects of surface reflection on solar radiances. These have been re-projected to the HRV

grid using bilinear interpolation. Within CPP and SICCS, the effects of atmospheric absorption by water vapor and ozone are accounted for, using the vertically integrated column values obtained either from the CAMS reanalysis or from the ECMWF forecast. In addition, aerosol properties are represented in the SICCS scheme by using either the CAMS reanalysis or CAMS



forecast as inputs. For this purpose, the aerosol optical depths at 468 nm and 865 nm are converted to a corresponding aerosol optical depth at 500 nm and the Ångström exponent, the input parameters expected by SICCS.

## 3.5 Use of the HRV Channel

A high-resolution cloud masking scheme is an essential prerequisite for increasing the spatial resolution of SEVIRI cloud
property retrievals, as uncertainties are expected to be largest for partly cloud-filled standard-resolution pixels. In particular, broken and inhomogeneous cloud fields are usually characterized by an abundance of such pixels (Werner et al., 2018). This implies that frequently only some of the 9 sub-pixels in a cloudy standard-resolution pixel are actually cloudy, leading to false positive detections. Also, the higher resolution improves the detection of small-scale cumuli, which might go undetected at the coarser spatial resolution of the narrow-band channels and result in false negative detections (Bley and Deneke, 2013).

The present study applies the threshold-based scheme introduced by Bley and Deneke (2013). Temporal averaging is used to obtain clear-sky reflectance composites for the HRV channel, while a regional threshold relative to the clear-sky composite is used to yield a binary classification of clear and cloudy observations. The Matthews Correlation Coefficient (MCC, Matthews, 1975) is used for threshold selection to quantify and maximize the accuracy of the resulting classification. Using a cloud mask obtained at standard resolution as first guess, an iterative algorithm is applied to update the clear-sky reflectance composite and
the threshold values, thereby optimizing the accuracy of the resulting HRV-based cloud mask. Details about the full algorithm, including the calculation of the HRV clear-sky composites and the selection of regional HRV reflectance thresholds can be found in Bley and Deneke (2013).

Compared to the original scheme, a number of updates have been made. First, the cloud mask from the NWC SAF v2016 software is now used as basis instead of the one provided by EUMETSAT's Meteorological Processing Facility. For derivation
of the clear-sky composite and thresholds, observations are aggregated over 16 days and 1 hour of observations (i.e. 12 RSS time slots), thus for a total number of $16 \times 12 = 192$ scenes. A temporal overlap of 8 days between successive analysis periods is used to ensure that the clear-sky composite can respond to rapid changes in surface reflectance. While the original scheme was based on regional thresholds, only three different thresholds for land, sea and coastal areas are used in the present scheme to improve the overall stability. This classification is based on the land-sea mask obtained from the GSHHS dataset, as described
in Sec. 3.4. Pixels containing both land and water in a neighborhood of $9 \times 9$ HRV pixels are considered as coastal/shore.

For the retrieval of cloud properties, a linear model is assumed following Cros et al. (2006) and Deneke and Roebeling (2010) to link the reflectances of the 0.6 μm, 0.8 μm, and HRV channels denoted by $r_{06}$, $r_{08}$, and $r_{H}$, respectively:

$$r_{H} = a\,r_{06} + b\,r_{08}. \tag{1}$$

For obtaining the coefficients $a$ and $b$, an empirical least-squares regression is applied. Prior to the regression, the HRV channel
images are first smoothed with the averaged MTF of the 0.6 μm and 0.8 μm channels, and are subsequently downsampled to the standard resolution. Fit coefficients are again calculated for reflectances aggregated over 16 days and 1 hour of observations. More discussion on the accuracy of this fit can be found in Cros et al. (2006) and Deneke and Roebeling (2010), as well as the analysis in Werner and Deneke (2020).





Within the modified CPP retrieval, Eq. 1 is not applied to the absolute values of reflectance, but only to their high-frequency residuals, which are defined as follows:

$$\delta r = \hat{r} - \tilde{r}. \tag{2}$$

Here, $\hat{r}$ denotes the actual reflectance field as observed with the HRV channel (i.e. sampled on the HRV grid and smoothed with the MTF of the HRV channel), while $\tilde{r}$ denotes the field sampled on the HRV grid but smoothed with the averaged MTFs of the $0.6\,\mu$m and $0.8\,\mu$m channels.

Solving for $\delta r_{06}$, the following relation is found:

$$\delta r_{06} = \frac{1}{a}(\delta r_{\mathrm{H}} - b\,\delta r_{08}). \tag{3}$$

As this equation contains $\delta r_{06}$ and $\delta r_{08}$ as unknowns, a further constraint is required. While an empirical relation obtained from coarser spatial scales was used by Deneke and Roebeling (2010), the lookup tables of the CPP retrieval are utilized here. An initial value of $\delta r_{06}$ is calculated from Eq. 1 by assuming that its value is equal to $\delta r_{08}$. In each retrieval iteration, a refined value of $\delta r_{08}$ is calculated by means of the following equation:

$$\delta r_{08} = \mathcal{F}_{08}(\hat{\tau}, r_e) - \mathcal{F}_{08}(\tilde{\tau}, r_e) \tag{4}$$

Here, $\mathcal{F}_\lambda$ denotes a function which utilizes the CPP lookup tables as forward model of the channel reflectances, in this case at the wavelength $\lambda = 0.8\,\mu$m for a given set of cloud properties and ancillary input variables. Thus, this step does not depend on actual observations at $0.8\,\mu$m, but only relies on the lookup tables and ancillary data of the retrieval.

While it is simplest to implement the cloud retrievals based on the assumption that $\delta r_{16} = 0$, i.e., that the high-frequency residual of the absorbing channel reflectance can be neglected, this assumption has been found to cause a degraded accuracy of the retrieved effective radius even in comparison to the standard resolution retrievals (see Werner and Deneke (2020) and the discussion in Sec. 4.1).

[Figure 4 about here.]

Instead, the retrieval has been modified to determine the high-frequency residual $\delta r_{16} = 0$ based on the tangent of the $\tau$-contour at the location of the standard-resolution reflectances in the Nakajima-King diagram (referred to as lookup-table approach with slope adjustment in Werner and Deneke (2020)). Mathematically, this can be expressed as the slope of the $\tau$-contour at the point $\mathcal{F}(\tilde{\tau}, \tilde{r}_e)$, or equivalently, as the ratio of the partial derivatives with respect to $\tau$ at that point:

$$\delta r_{16} = \frac{\frac{\partial}{\partial \tau} \mathcal{F}_{16}(\tilde{\tau}, \tilde{r}_e)}{\frac{\partial}{\partial \tau} \mathcal{F}_{06}(\tilde{\tau}, \tilde{r}_e)} \delta r_{06}. \tag{5}$$

This approach is illustrated in Fig. 4. For more details, readers are referred to the companion paper of Werner and Deneke (2020).



## 4 Application Examples

This section presents three application examples for the cloud products from the improved SEVIRI retrieval scheme. The examples have been selected to demonstrate the benefits that can be gained from the increase in spatial resolution, and are compared to results obtained at SEVIRI's standard resolution.

### 4.1 Shallow Convective Clouds

[Figure 5 about here.]

[Figure 6 about here.]

The main motivation for the development of the HRV-based cloud retrieval scheme has been the expectation that the increase in spatial resolution will lead to more accurate cloud retrievals, and will bring the instrumental capabilities of SEVIRI closer to those of MODIS. Improvements are expected to be significant in particular for shallow convective clouds due to their comparatively small size and their large spatio-temporal variability.

To verify this aspect, a shallow convective cloud field is considered here, and retrieval results are contrasted to those obtained from collocated MODIS observations. A scene viewed by the MODIS instrument flown aboard the Terra Earth observing satellite on 2 June 2013 at 10:50Z over North-Eastern France has been selected for this purpose. The choice of observations from Terra allows the consistent use of MODIS retrievals based on the $1.6\,\mu\mathrm{m}$ channel for comparison with SEVIRI, as this channel of the MODIS instrument is affected by defective detectors on Aqua. The MOD06 cloud properties from the collection 6.1 release are used here, and retrieval results for fully overcast and partially cloudy pixels have been combined. It should be realized that in contrast to the results presented in Werner and Deneke (2020), products from two inependent retrievals and two different instruments are compared, thus deviations are expected to be substantially larger than the results presented in that study.

Fig. 5 shows the fields of $\tau$ obtained for the example scene provided by MODIS, and both the standard and improved HRV-based SEVIRI retrievals. SEVIRI data has been re-projected to the MODIS grid using nearest-neighbour interpolation, and a translation has been applied to account for parallax shift and cloud motion in combination with a mismatch in observation time of about one minute. This translation has been determined by maximizing the cross-correlation of both $\tau$-fields, and results in a shift of the SEVIRI data by about $2.6\,\mathrm{km}$ and $0.4\,\mathrm{km}$ in North and East directions, respectively. It is clearly evident that the increased spatial resolution obtained by using the HRV channel in the retrieval helps to better resolve the small-scale structure of this cloud field. This visual impression is confirmed quantitatively by a significantly higher correlation coefficient of about 0.78 found for the HRV-based $\tau$ field and the corresponding MODIS C6.1 product, compared to a value of 0.47 obtained for the standard-resolution retrieval results.

Fig. 6 shows the corresponding histograms of the derived $\tau$ using logarithmic bin spacing for this scene. The standard-resolution SEVIRI retrieval exhibits the narrowest distribution of values, with too few optically thin and thick clouds compared to the MODIS product. While the HRV-based SEVIRI retrieval still yields fewer optically thick clouds than MODIS, it reports





a similar amount of optically thin clouds, and is able to better reproduce the dynamic range of the MODIS product than the standard-resolution retrieval scheme. For the standard retrieval, the maximum value of retrieved $\tau$ is only 16.5, while values of 40.3 and 61.8 are observed for the SEVIRI HRV-based and MODIS products, respectively. A likely explanation for the remaining underestimation is the oblique viewing angle of Meteosat over Europe, which increases the pixel size in North-South

direction by a factor of about 2, in combination with the lower optical resolution of SEVIRI, and limits the maximum $\tau$ for the HRV-based retrieval below that of MODIS. The HRV-based retrieval also reports a significantly larger number of optically thin clouds compared to MODIS. While it is beyond the scope of this article to fully resolve the remaining discrepancies, they are likely due to differences in retrieval algorithms, sensor calibration, and/or viewing geometry. In particular, the MODIS processing scheme has a rather strict quality control, which might be responsible for the fact that no values are being reported

for these rather optically thin clouds, despite our choice to also include MODIS results for partially cloudy pixels.

It should be noted that for solar energy applications, the correct representation of $\tau$-values at and below a value of 5 is highly relevant, as such values will result in non-zero direct irradiance. While rejecting such retrieval results in the cloud retrieval scheme due to their large uncertainties will most likely improve the $\tau$-retrieval accuracy itself, it will cause a subsequent overestimate of SSI if these pixels are assumed to be cloud-free. Both global and direct irradiance components will be affected,

but errors will be most pronounced for the direct irradiance and the direct-diffuse ratio, parameters which are critical for the calculation of the tilted irradiance, e.g., on the plane of a photovoltaic module or the focal plane of a concentrating solar power plant.

For the effective radius, no significant improvement is found resulting from the use of the HRV channel in the retrieval, and correlations between SEVIRI and MODIS results are relatively low for this scene. Restricting the comparison to pixels

with $\tau$ exceeding a limit of 6 for both MODIS and SEVIRI to ensure reliable effective radius retrievals, Pearson correlation coefficients of 0.43 and 0.42 are found for the HRV and standard-resolution effective radius results, respectively. The reader is reminded here that a similar magnitude of the correlation is expected, as the retrieval constraint ensures that the effective radius is close to that of the standard-resolution retrieval in the iterative algorithm. In consequence, a comparatively high correlation coefficient of 0.85 is found between the two SEVIRI retrievals at the different spatial resolutions. A modification of the retrieval

to only use the smoothly interpolated value of the 1.6 $\mu$m reflectance instead results in a sharp reduction of the correlation of the high-resolution retrieval results with the MODIS $r_e$ to a negative value of -0.05. This finding emphasizes that despite the seemingly low values of correlation for $r_e$ found above, the choice of the retrieval constraint is important to ensure that the accuracy of the standard-resolution $r_e$ is not degraded by use of the HRV channel.

## 4.2   Detection of Convection Initiation

[Figure 7 about here.]

[Figure 8 about here.]

An application which is expected to benefit from the improved high-resolution SEVIRI cloud property retrievals is the early detection of convective initiation (CI). Current approaches for the SEVIRI instrument are mainly based on the narrowband



channels, and are thus limited to the standard spatial resolution of SEVIRI (e.g. Mecikalski et al., 2010). Some previous studies already point out some benefits arising from the use of the HRV channel (e.g. Carbajal Henken et al., 2011; Mecikalski et al., 2013; Merk and Zinner, 2013). Its higher spatial resolution could thus be one way to improve the lead time for the detection of convective initiation in current MSG-based CI detection schemes. In particular, developing convective clouds can

likely be resolved earlier in their life cycle, and small-scale variability of convective clouds can be better resolved.

To illustrate this aspect, an example case considering the early growth phase of a deep convective cloud system which formed on 18 June 2013 is presented here. On this day, a series of orographically triggered thunderstorms developed along the mountain range of the Thuringian forest. The weather over Central Europe was dominated by an upper-level high pressure ridge, with an extension from Northern Africa to South Western and Central Europe. An upper-level low pressure system was

located over the Iberian peninsula, and advected hot and unstable air masses northward towards Central Europe on the forward flank of its frontal zone. This led to an unstable atmospheric situation with extremely high values of mixed-layer Convective Available Potential Energy (CAPE) reaching up to $3600 \, \mathrm{J \, kg^{-1}}$ over Southern and Central Germany, as diagnosed from the Global Forecast System reanalysis [4]. The first shallow convective clouds started to form around 10:30Z along the Thuringian Forest.

The convective cell considered here initiated at a location and time of 11°16'57" E and 50°29'30" N and 13:50Z, developed into a cold ring storm (Setvák et al., 2010), and lasted for more than five hours, before dissipating around 19:30Z. An increase of radar reflectivity above a threshold of $35 \, \mathrm{dBZ}$ has been adopted to determine the location and timing of CI for this cell (Senf and Deneke, 2017), based on observations from the German weather radar network.

Fig. 7 shows a time sequence of frames of the *day-natural color* RGB composite based on SEVIRI's 1.6 0.8 and 0.6 $\mu$m

reflectances and downscaled to HRV resolution, overlaid with contours of the 10.8 $\mu$m brightness temperature for a time period from 13:30Z to 14:10Z and centered on the location of CI. At 13:30Z, thus 20 minutes prior to CI, the cell started to grow rapidly in vertical extent. Beginning as a rather small and shallow cloud with a minimum cloud top temperature around the freezing level, it reached $260 \, \mathrm{K}$ five minutes later, and values below $240 \, \mathrm{K}$ at 13:45Z.

In the following analysis, pixels belonging to the convective cloud object of interest have been identified using an adaptive

threshold in 10.8-$\mu$m brightness temperature. For this purpose, an object mask has been constructed using a threshold which is 5–20 K warmer than the observed minimum brightness temperature, increasing the spread linearly from 5 K at 273.15 K to 20 K at 220 K, and keeping it constant above or below this range. Fig. 8 shows a comparison of the temporal evolution of $\tau$ for the convective cell based on results from the improved high-resolution and standard-resolution retrievals. For this purpose, the standard-resolution $\tau$-retrieval has been interpolated to the HRV grid using bilinear interpolation for consistency. Already in

the early stages of the cloud life cycle starting at 13:20Z, some pixels show optically thick $\tau$-values exceeding 50 for the high-resolution retrieval. In contrast, maximum values of $\tau$ remain below 20 up to 13:40Z for the standard-resolution retrieval. A notable increase in the median of $\tau$ can be seen at 13:30Z and 13:40Z for high- and standard-resolution $\tau$-results, respectively. This implies that at least for the considered convective cell, growth signatures in $\tau$ are found about 10–20 minutes earlier for the high-resolution retrievals depending on the considered quantity, potentially increasing the lead-time for the detection of CI.

---

[4]Diagnosed based on analysis maps from http://www1.wetter3.de/archiv_gfs_dt.html, last access: 3 August 2020.





## 4.3 Surface Solar Irradiance

[Figure 9 about here.]

[Figure 10 about here.]

[Figure 11 about here.]

In this subsection, benefits of the increased spatial resolution of the improved retrieval scheme for estimating SSI are discussed. In climate studies, it is usually sufficient to average radiative fluxes over longer time periods (e.g. Wild, 2020). In contrast, various authors (e.g. Wiemken et al., 2001; Perpiñán et al., 2013; Lave et al., 2015) have demonstrated the need to quantify irradiance variability in the 1 minute or even 1 second range, in order to capture the full range of natural fluctuations relevant for the grid integration of solar power generation.

High frequency variability is introduced by clouds with sizes smaller than the pixel resolution, small-scale cloud heterogeneity and 3-dimensional (3D) radiative transfer effects, and can even result in enhanced irradiances (Schade et al., 2007). Such effects are poorly represented in satellite-based SSI products due to their reliance on 1D radiative transfer (Deneke et al., 2005), and due to under-sampling in space and time by the satellite observations. In particular, the high correlation of TOA and surface fluxes implied by 1D radiative transfer breaks down for 3D radiative transfer (Kassianov et al., 2005), resulting in a decorrelation of atmospheric transmission and reflection as observed by ground-based and satellite observations (Deneke et al., 2009). It is of high scientific relevance which part of the SSI variability is missed by satellite retrievals and how this is affected by the spatial and temporal resolution of satellite observations. In the following case study, it is thus investigated how the increase in spatial resolution of the improved HRV-based retrieval changes the agreement of satellite-based SSI with surface observations.

As reference, a unique dataset of observations from a dense pyranometer network operated during the High Definition Clouds and Precipitation for advancing Climate Prediction (HD(CP)$^2$) Observational Prototype Experiment (HOPE) is used here. HOPE was a field experiment that took place in Jülich, Germany, from 3 April to 31 July 2013 (Macke et al., 2017). It was conducted to provide a broad range of observational datasets for the evaluation of the atmospheric icosahedral non-hydrostatic (ICON) model developed within the HD(CP)$^2$ project (Heinze et al., 2017). Each of these stations was equipped

with a silicon photodiode pyranometer (model: EKO ML-020VM) to measure the global horizontal irradiance at 10 Hz resolution, a micromodule to measure air temperature and relative humidity (Driesen+Kern DKRF 4001P), and an embedded Global Positioning System (GPS) module as an accurate time reference. A description of the network and the resulting dataset is given in Madhavan et al. (2016). An overview of the distribution of stations is shown in Fig. 9. As the stations were placed within an area of only $8 \times 10 \, \text{km}^2$, many of the inter-station distances were smaller than the SEVIRI pixel resolution even for its HRV

channel. This unique dataset can thus provide novel insights into the small-scale variability of global irradiance unresolved by current geostationary satellite observations. Madhavan et al. (2017) introduced methods to estimate the deviation between a point measurement and the spatial average for a surrounding domain, and how spatial averaging affects the power spectrum of the corresponding SSI time series. Due to the large number of pyranometers, robust evaluation statistics can be obtained even





for single days. 24 May 2013 has been chosen here as a case study. The weather on this day was influenced by an upper-level trough in combination with a North-Westerly air flow, advecting polar airmasses to Central Europe and leading to convective activity and showers. The occurence of Cumulus congestus together with relatively strong winds caused significant variability in the SSI.

For each station, a matching satellite-based time series has been extracted from the standard- and HRV-resolution satellite products , weighting pixels by a Gaussian centered on the station location with a standard deviation of $1\,\mathrm{km}$ in North-South and East-West directions, respectively, and accounting for the distorted shape of satellite pixels due to the oblique viewing geometry. A shift in collocation has been applied here to maximize the daily correlation between ground-based and satellite time series, having a magnitude of 2700 meters and 3000 meters in West and North directions, respectively. A time period from

8:00Z in the morning until 16:00Z in the afternoon has been considered, and the time series of the 63 stations that passed the quality control criteria described by Madhavan et al. (2016) are included in the analysis. The SSI time series have been smoothed using a 5 minute running mean before sub-sampling to the observation time of SEVIRI, accounting for the delay between start of scan and acquisition time for the considered region.

     Fig. 10 shows the median time series together with the inter-quartile range ($25^{\mathrm{th}}$–$75^{\mathrm{th}}$ percentile) and the $5^{\mathrm{th}}$ – $95^{\mathrm{th}}$ per-

centile range obtained from the pyranometer observations and the two satellite-based retrievals. The statistics for both satellite datasets and the pyranometer observations are calculated consistently based on the time series obtained at the location of the pyranometer stations. High spatio-temporal variability is observed, as indicated by the large range of values at specific times as well as large temporal changes throughout the day. The pyranometer-based dataset (black) clearly exhibits the largest spatio-temporal variability, while the HRV-based product (red) also shows notably more variability than the standard-resolution

product (blue). It also seems to capture the variability of the pyranometer dataset better. Specifically, minima and maxima in the pyranometer record appear to be better resolved. In particular, the standard-resolution SSI data show an overestimate for thicker clouds, consistent with the underestimation of $\tau$-values for optically thicker clouds already discussed in Sec. 4.1 and Fig. 6.

     To support this visual impression, Fig. 11 displays the root-mean square error (RMSE) for the two satellite-based products

using the pyranometer irradiance as reference. The RMSE is derived for several averaging periods between 5 minutes and 1 hour. A strong reduction of RMSE with averaging period is observed: while for 5 minute averages, RMSE values of $184\,\mathrm{W}$ $\mathrm{m^{-2}}$ and $200\,\mathrm{W\,m^{-2}}$ are found, this reduces to $58\,\mathrm{W\,m^{-2}}$ and $64\,\mathrm{W\,m^{-2}}$, respectively, for hourly averages. Regardless of averaging period, a reduction of the median RMSE by about 10% is found for the HRV-based product. Applying Mood's median test (Mood, 1950) to test the difference of the distributions, a reduction is found for the RMSE which is statistically

significant at the 95 % confidence level for all but the 30 minute averaging period.

     It is worth pointing out that the inter-quartile ranges of the RMSE for different stations and averaging periods have a magnitude of about $20 - 30\,\mathrm{W\,m^{-2}}$, and that the distributions partly overlap for the two satellite-based SSI products. Hence without the large number of stations of the pyranometer network, it would not have been possible to diagnose the accuracy improvement achieved by the HRV-based SICCS product for this case study with any statistical confidence. While additional

days or observations across a larger region could have been used, this complicates a comparison by combining observations





with different cloud types and synoptic conditions, both factors likely to influence the absolute product accuracy and the sensitivity to spatial resolution. A further interesting aspect is the sensitivity of the RMSE to the applied shift in collocation, which is particularly strong for the HRV-based retrieval. While we have empirically chosen an optimal collocation here, this raises the question how to collocate ground-based and satellite observations in an operational setting, a aspect which becomes

more important with increased spatial resolution.

## 5  Conclusions and Outlook

Within the present paper, it has been demonstrated that it is possible to improve the spatial resolution of MSG SEVIRI-based cloud property and subsequent SSI retrievals by use of its HRV channel. For this purpose, the HRV reflectance is first used in a threshold-based cloud mask, while the high-frequency component of the HRV reflectance is subsequently extracted with a

high-pass filter and utilized as physical constraint to resolve small-scale variability in cloud optical depth. As no information is available on the small-scale variability of effective radius, a constraint based on the lookup tables used by the cloud property retrieval has been implemented in the present version of the algorithm.

An overview of the complete retrieval scheme has been given here, including a description of the modifications made to the CPP retrieval to utilize the HRV reflectances for improving the spatial resolution, the ancillary data incorporated in the

scheme to optimally benefit from the improved spatial resolution, and some other changes to improve the base retrieval. A more in-depth evaluation of the applied downscaling algorithm, an evaluation of its accuracy based on MODIS observations, and a discussion of the choice of the constraint imposed for the effective radius can be found in the companion paper by Werner and Deneke (2020).

Three applications of the resulting cloud and radiation products have been presented to highlight some benefits arising from

the improved spatial resolution. First, operational MODIS C6.1 and SEVIRI retrievals of $\tau$ and $r_e$ have been compared for a shallow convective cloud field, showing better agreement and enhanced capabilities to resolve the small-scale variability of $\tau$. The initiation and growth phase of a severe convective storm has been analyzed, indicating that the formation of an optically thick cloud corresponding to the growing convective cell can be recognized 10–20 minutes earlier. This case study shows promise for improving the detection of convection initialization in the future. Finally, the retrieved time series of SSI has been

compared with observations from a dense pyranometer network, showing a statistically robust improvement in the agreement of the satellite retrievals with the ground-based observations, leading to a reduction of the RMSE by about $10\%$.

Future work in several directions is warranted. While the present version of the retrieval algorithm already seems to yield results superior to those of the standard-resolution scheme, we do believe that there are several aspects of the algorithm that can be further improved. This includes the treatment of small-scale variability in effective radius, either by using physical

constraints such as the adiabatic model (see e.g. Merk et al., 2016), or the use of empirical parameterizations, possibly adapted to the local cloud type. The use of corrections for partially cloud-filled pixels, such as those proposed by Werner et al. (2018) are also expected to be beneficial. Complementary ideas for using the HRV channel to correct infrared brightness temperatures



for effects of partially cloudy pixels and to improve estimates of cloud top temperature have been presented by Mecikalski et al. (2013).

Similar approaches could also be adopted to other meteorological multi-resolution imagers such as the Advanced Baseline Imager aboard the current generation of geostationary GOES satellites, or the polar-orbiting MODIS and VIIRS sensors, and to

increase the spatial resolution of the cloud and radiation products up to that offered by the highest-spatial resolution channels. It has however to be cautioned that a higher spatial resolution does not necessarily imply a higher product accuracy. Specifically, the findings of Zinner and Mayer (2006) suggest that at resolutions higher than $1\,\mathrm{km}$, three-dimensional effects become more prominent, causing increasing deviations from the assumption of 1D radiative transfer underlying current retrievals. In addition, the use of other absorbing wavelength reflectances in the cloud property retrieval, e.g. at $2.2\,\mu\mathrm{m}$ might influence the accuracy

of the downscaling algorithm.

Despite these caveats, the comparison or even synergistic combination of satellite and ground-based observations critically depemnds on the collocation of the observations in terms of the sampled atmospheric volume. At least in this respect, an improved spatial resolution of the satellite products is likely always beneficial, due to the fact that most ground-based observations are able to resolve much finer-scale cloud structures.

In future work, a more extensive evaluation of the resulting product quality should be conducted. Specifically, the results presented here for the three example application should be extended to a larger number of satellite scenes in order to confirm the representativeness of the findings. A number of additional products are derived from satellite observations based on $\tau$ and $r_e$. Examples are the cloud liquid water path and droplet number concentration, as already considered in Werner and Deneke (2020), or the SEVIRI-based estimation of rain rate (see e.g. Roebeling and Holleman, 2009).

Concrete plans exist in particular to further investigate the dependence of the accuracy of SSI retrievals on the spatial and temporal resolution of the satellite data, due to the relevance of satellite-based SSI products for solar energy applications. Specifically, a comprehensive evaluation against data from the pyranometer network is planned, including the MetPVNet measurement campaigns in autumn 2018 and summer 2019, which cover a different region and different cloud conditions. Meilinger et al. (2020) find deviations of up to $\pm600\,\mathrm{W/m^2}$ depending on cloud type when comparing CAMS satellite products

and ground measurements of solar irradiance with a temporal resolution of 1 minute for the first of these campaigns.

Finally, while parts of the benefits of the presented scheme will be available in the future by use of observations from the METEOSAT Third Generation (MTG) Flexible Combined Imager, climate applications often depend on the availability of long-term records. Hence, even after launch of MTG, the SEVIRI-based scheme might present a pathway toward creating a more homogeneous long-term climate data record of cloud properties at high spatial resolution, based on both MTG and MSG

observations.

*Code and data availability.* The datasets used for the analyses presented in this paper, and the Python codes used for preparing the CPP input and paper figures are available from the first author on request, and will be made publically available through the ZENODO data repository



for the final paper. The CPP retrieval software is copyrighted by EUMETSAT and is not publically available. The NWC SAF software is available to registered users based on conditions given at: http://www.nwcsaf.org/. LSA SAF products are available CAMS Reanalysis data

*Author contributions.* Conceptualization: H.D.; Methodology: H.D., S.B., F.W., M.S.H.; Software: H.D., S.B., P.W., J.F.M., F.S., F.W.; Formal analysis: H.D., C.B.V. (Sec. 4.3), S.L. (Sec. 4.2), M.S.H., F.S. (Sec. 4.2), J.W.(Sec 4.3), F.W.; Writing - original draft preparation, H.D.,
C.B.V. (Sec. 4.3), S.L. (Sec. 4.2), J.F.M. (Sec. 3.2), M.S.H., F.S. (Sec. 3.1 and 4.2), J.W. (Sec 4.3), F.W.; Writing- review and editing: all authors; Funding Acquisition: H.D., A.H., A.M., M.S.H.; All authors have read and agreed to the submitted version of the manuscript.

*Competing interests.* The authors declare no competing interests.

*Acknowledgements.* The authors thank Stefanie Meilinger for initiating the MetPVNet project, and for her valuable comments on an earlier version of this manuscript. The initial concept for this investigation was conceived during a EUMETSAT research fellowship of the first au-
thor hosted at KNMI. The associated financial support and discussions with EUMETSAT and KNMI colleagues is greatly appreciated. This study was carried out within the frame of the German collaborative project MetPVNet funded by the German Ministry of Commerce, grant numbers 0350009E. The use of the following data and software is gratefully acknowledged: SEVIRI satellite data distributed by EUMETSAT and obtained from the TROPOS satellite archive; MODIS data obtained from the Level-1 and Atmosphere Archive and Distribution System (LAADS) Distributed Active Archive Center (DAAC); surface albedo data generated and distributed by the LSA SAF; CAMS Re-
analysis data obtained from the Copernicus Atmosphere Monitoring Service; the GSHHG dataset developed and maintained by P. Wessel (SOEST, University of Hawaii, Honolulu, HI) and W. Smith (NOAA Geosciences Lab, National Ocean Service, Silver Spring, MD); the SRTM15_PLUS topography provided by the Satellite Geodesy research group at the Scripps Institution of Oceanography; the NWC/GEO software package developed and distributed by the NWC SAF; the CPP software developed by KNMI as part of the CM SAF; various open source software packages used for this work, including the Python programming language including various SciPy packages. Results contain
modified Copernicus Atmosphere Monitoring Service information (2020).



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



## List of Figures









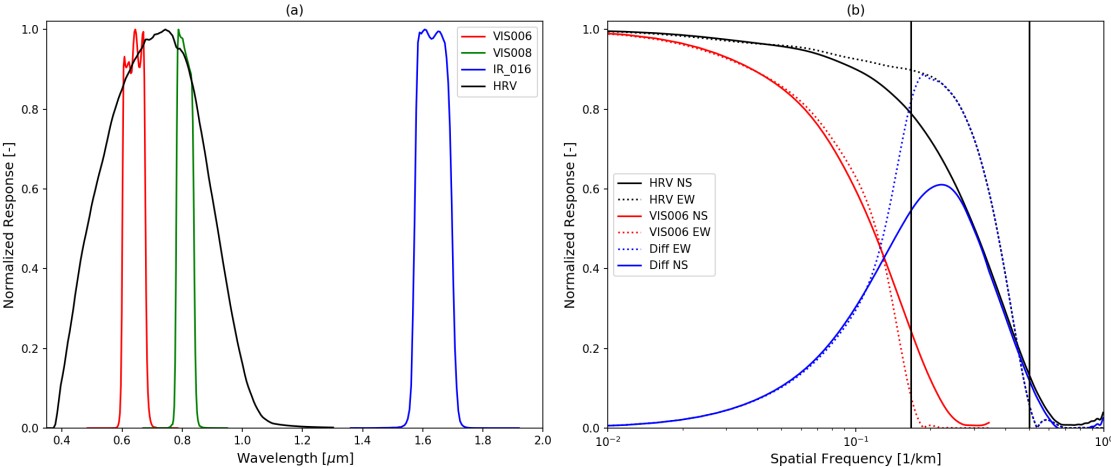

**Figure 1.** Normalized spectral response of the solar channels of the Meteosat SEVIRI instrument (a). The red, green, blue and black lines correspond to the 0.6 $\mu$m (VIS006), 0.8 $\mu$m (VIS008), 1.6 $\mu$m (IR_016) and HRV channels, respectively. Modulation transfer function of the HRV (black) and the 0.6 $\mu$m (red) channels in North-South (NS, solid) and East-West (EW, dotted) directions (b). The vertical black lines show the Nyquist frequency limit of the respective channels depending on their sampling resolution. The differences (Diff) of the two modulation transfer functions in both directions are shown in blue. (adapted from Deneke and Roebeling (2010))


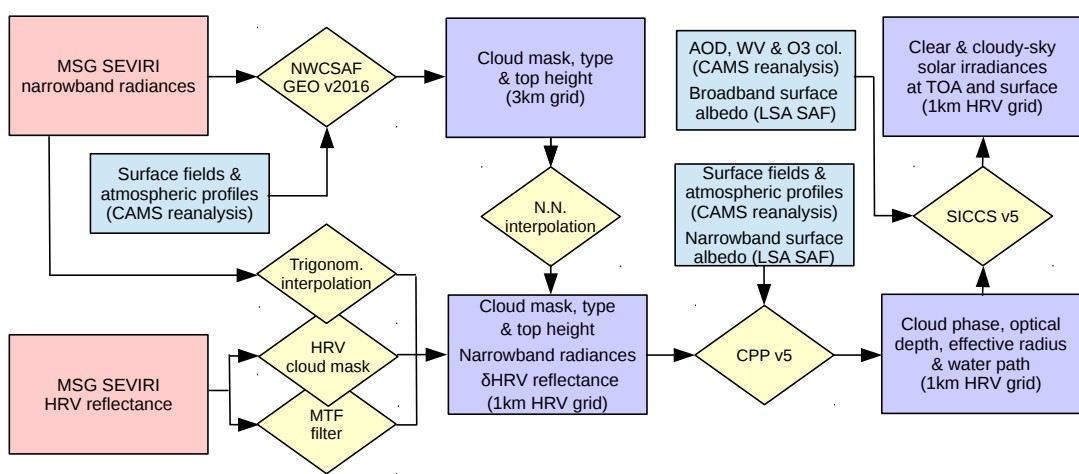

**Figure 2.** Illustration of the workflow of the improved cloud property retrieval scheme based on observations from the Meteosat SEVIRI instrument at its HRV channel resolution. See Section 3 for acronym definitions and further explanation.





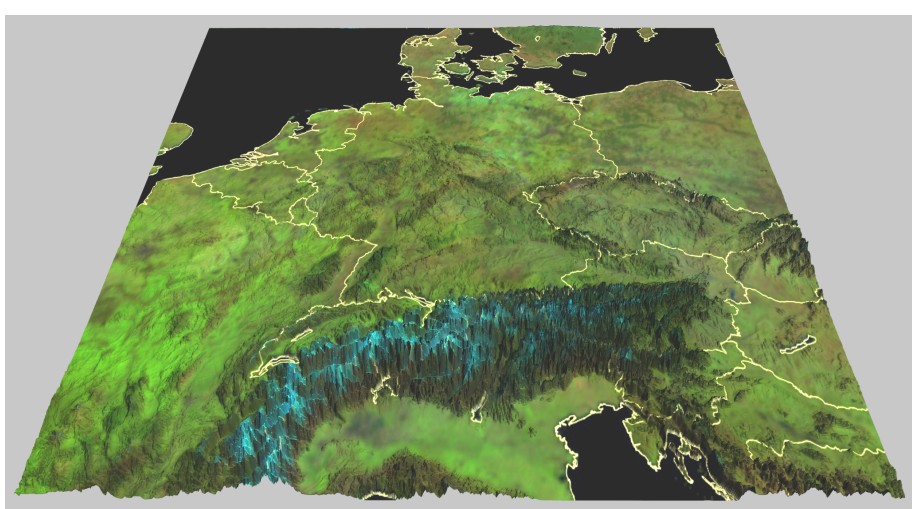

**Figure 3.** *Day-natural color* RGB (red green blue) false-color composite, based on the surface reflectance product of the LSA SAF at wavelengths of 1.6 $\mu$m, 0.8 $\mu$m and 0.6 $\mu$m used by the improved retrieval scheme. The RGB texture has been overlayed on a 3-dimensional perspective rendering of the elevation dataset used by the retrieval, which is derived from the SRTM15_PLUS digital elevation model. Country borders are shown as yellow lines.



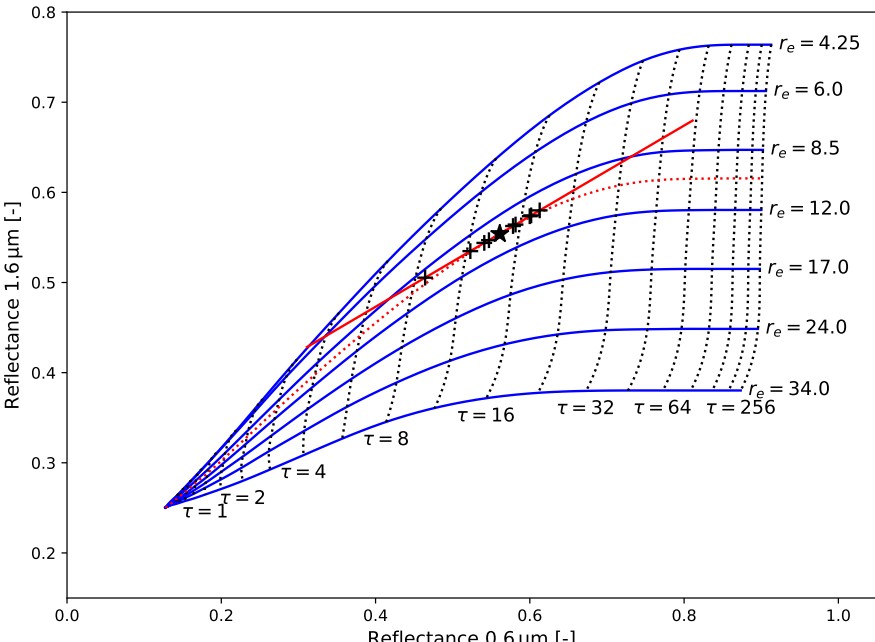

**Figure 4.** Illustration of the lookup table-based downscaling method with slope adjustment used in the improved HRV-based retrieval. The relation of the reflectances at wavelengths of $0.6\,\mu m$ and $1.6\,\mu m$ is shown for contours of pre-calculated cloud optical depth $\tau$ (dotted black lines) and effective radius $r_e$ (solid blue lines), based on detailed radiative transfer calculations. A hypothetical standard-resolution observation with $\tau = 14.0$ and $r_e = 10.0\,\mu m$ is marked by the black star. High-resolution retrieval results are marked by black crosses, and are constrained to fall on the red line, which is given as tangent to the $r_e$-contour at the point of the standard-resolution observation. See text for further explanation.





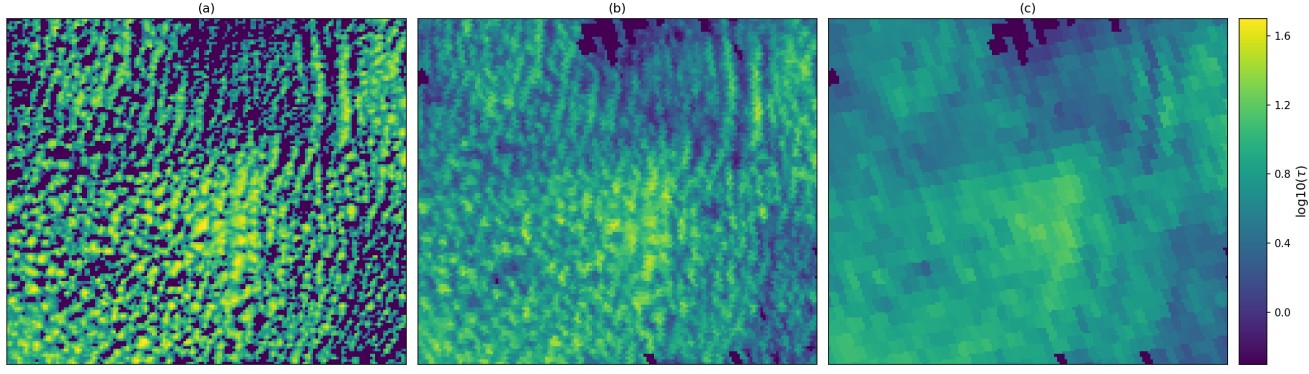

**Figure 5.** Cloud optical depth ($\tau$) of a shallow convective cloud field observed over North-Eastern France at $3°25'$ E and $48°7'$ N, on 2 June 2013 at 10:50Z. A logarithmic color scale is used. Values are shown for the operational Terra MODIS C6.1 retrieval (a), the improved Meteosat SEVIRI retrieval (b), and the standard-resolution Meteosat SEVIRI retrieval (c).



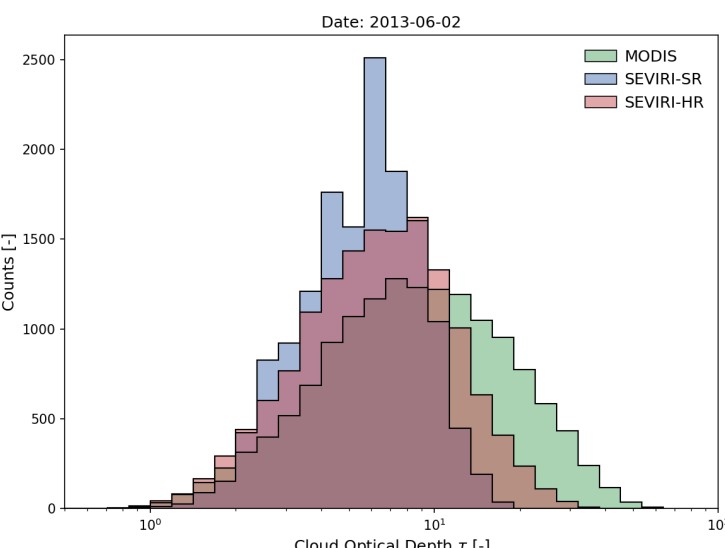

**Figure 6.** Histogram of cloud optical depth ($\tau$) using logarithmic bin spacing, for the cloud field displayed in Fig. 5. Values are shown for the Terra MODIS C6.1 retrievals (MODIS, green color), the improved HRV-resolution Meteosat SEVIRI retrieval (SEVIRI-HR, red color), and the standard-resolution SEVIRI retrieval (SEVIRI-SR, blue color).



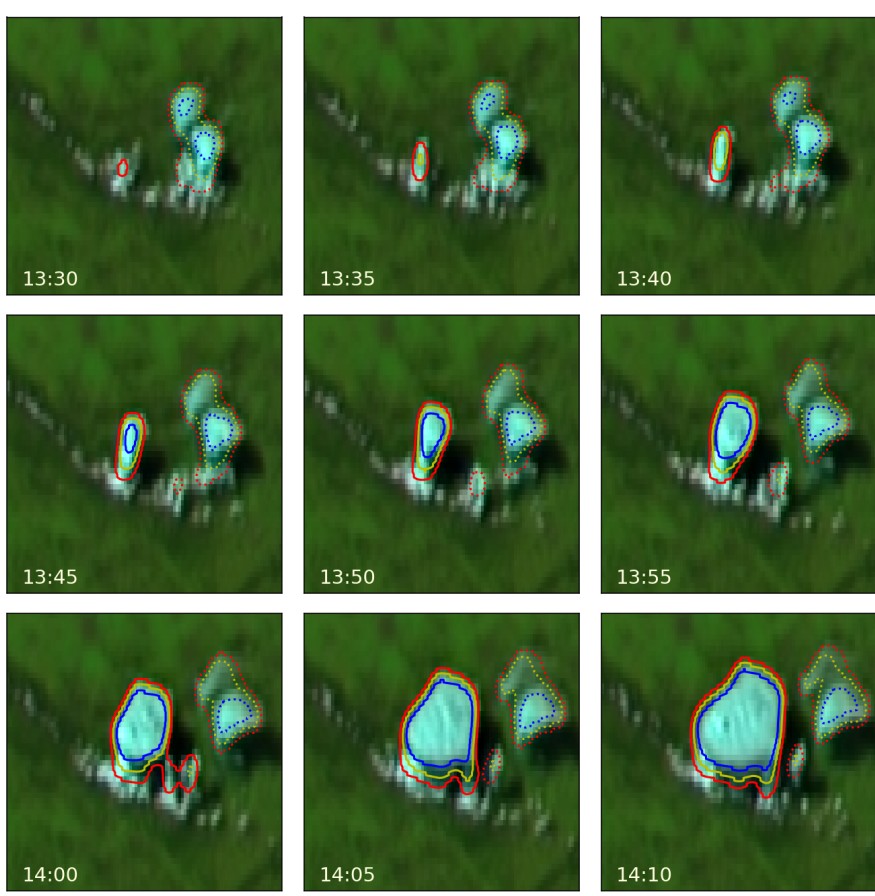

**Figure 7.** Time sequence of the *day natural color* RGB composite showing the SEVIRI reflectances for the 1.6, 0.8, and 0.6-$\mu$m channels, and downscaled to HRV channel resolution using the constant-reflectance ratio approach described in Werner and Deneke (2020). Frames correspond to a time period from 13:30Z – 14:10Z on 18 June 2013, and a region of $51 \times 101$ pixels (about $100 \times 100 \, km^2$) centered on the location of convective initiation occuring at $11°16'57"$ E and $50°29'30"$ N. An aspect ratio of 2:1 is used to compensate for the lower pixel resolution in North-South direction. Also shown are contours of the 10.8-$\mu$m brightness temperature (red=273.15 K; yellow=260 K; blue=240 K). Solid lines are used for the convective cell of interest, while dotted lines are used for other cells.



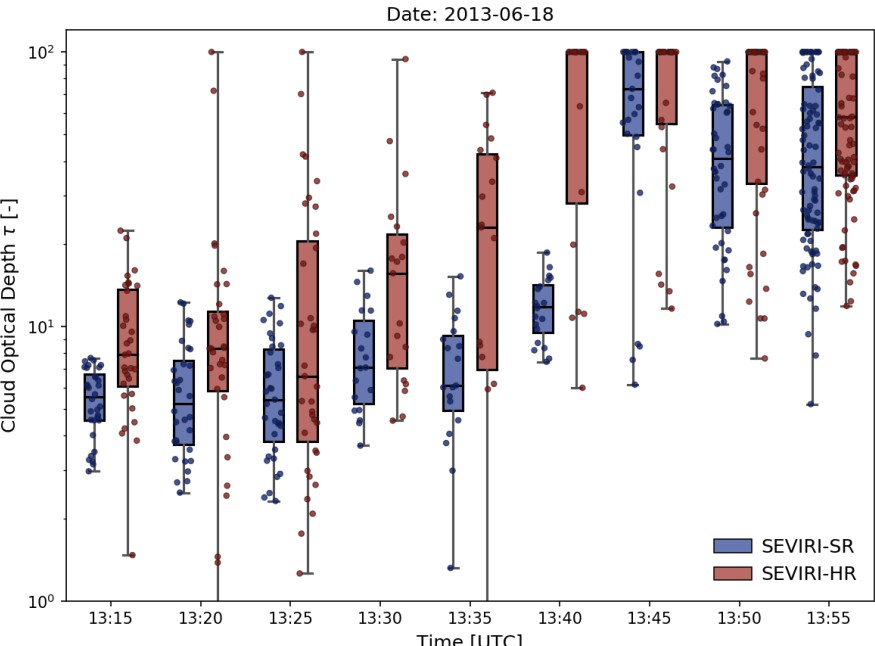

**Figure 8.** Box and whisker plot of cloud optical depth ($\tau$) as function of time for the convective cell shown in Fig. 7, comparing the standard-resolution retrieval (SEVIRI-SR, blue) and the improved HRV-based retrieval (SEVIRI-HR, red). A time period from 13:15Z – 13:55Z has been chosen to capture the early growth phase of the convective cell. The boxes extend from the lower to the upper quartile, the lines within the boxes mark the median, and the whiskers extend to the minimum and maximum values of $\tau$ for all pixels belonging to the cloud object. Aditionally, points have been added displaying the individual $\tau$ values of the object pixels.





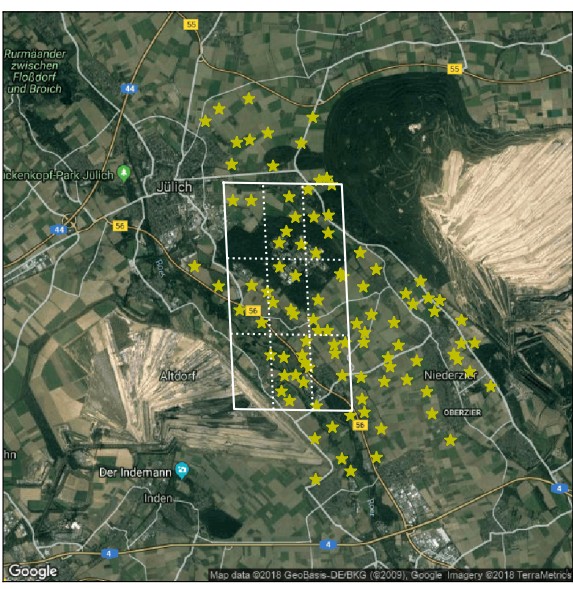

**Figure 9.** Google Maps™ satellite view showing the measurement region of the pyranometer network operated during the HOPE campaign in the vicinity of Juelich, Germany, in spring and summer of 2013. The individual pyranometer stations are marked by yellow stars, the location of the corresponding standard resolution MSG SEVIRI pixel is shown by solid white lines, and the $3 \times 3$ HRV pixels laying within this pixel are delimited by the dotted lines. Map data: ©Google, GeoBasis and TerraMetrics.



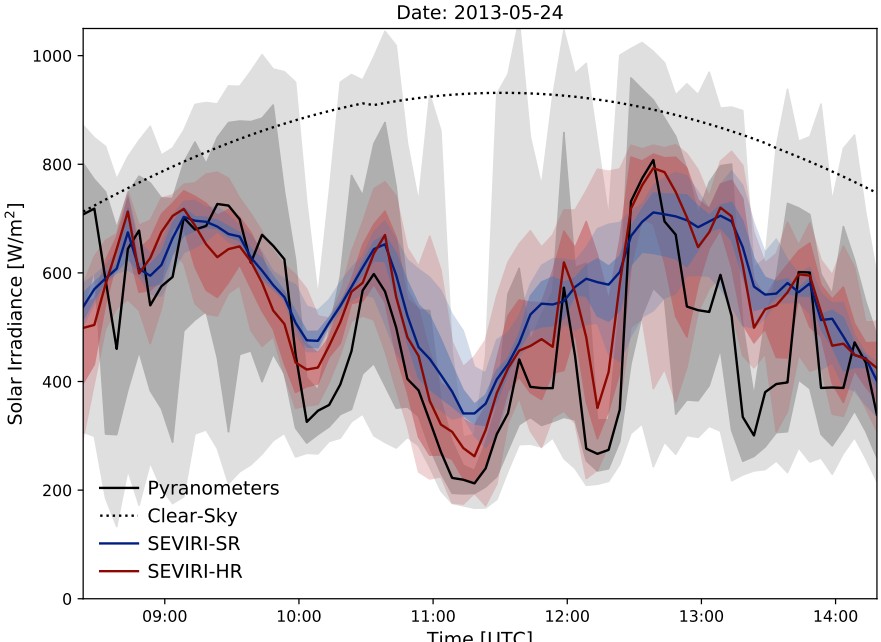

**Figure 10.** Time series of ground-based and SEVIRI-retrieved surface solar irradiance. Median (thick line), inter-quartile range (dark shading) and 5. – 95. percentile range (light shading) of SSI reported by the 63 pyranometers at the stations shown in Fig. 9 (black color), visualizing the spatial variability across the stations. The corresponding time series obtained from the standard-resolution (SEVIRI-SR) and the improved HRV-resolution (SEVIRI-HR) SEVIRI retrievals at the station locations are displayed in identical style using blue and red colors, respectively. All time series have been resampled to the 5-minute time resolution of the SEVIRI observations. The mean clear-sky irradiance from the satellite retrieval is shown as dotted black line.



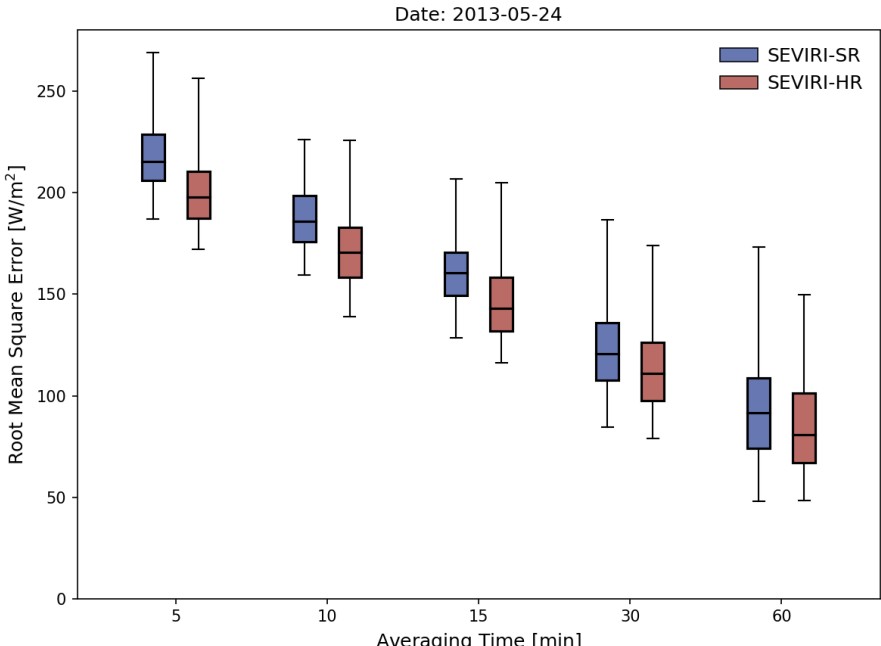

**Figure 11.** Distribution of the daily-mean root mean square error (RMSE) found from the comparison of pyranometer-based and satellite-derived SSI time series for the 63 pyranometer stations for 24 May 2013. A box and whisker plot is used for visualization, and averaging periods ranging from 5 minutes to 1 hours are considered. Boxes extend from the lower to the upper quartile, the line within the boxes displays the median, and the whiskers extend to the minimum and maximum values. Results are shown for standard-resolution (SEVIRI-SR) and high-resolution (SEVIRI-HR) SSI retrievals in blue and red colors, respectively.