# Peer review of "Increasing the spatial resolution of cloud property retrievals from Meteosat SEVIRI by use of its high-resolution visible channel: implementation and examples"

_Atmospheric Measurement Techniques, 2020_

## Referee Comment (RC1) · Anonymous Referee #1 · 2 Dec 2020

Summary:

My operational satellite seniors, such as MODIS, VIIRS, SEVIRI and ABI, have different spatial resolution for different spectral channels. For example, the SEVIRI has a 1 km x 1km (nadir) high-resolution visible (HRV) band, in addition to other spectral channels at 3 km x 3 km. This paper presents a method, partly based on Werner and Deneke (2020), to combine the HRV with other low-resolution bands to achieve cloud property retrievals at 1 km x 1km. The paper documents the algorithm implementations, presents a comparison with collocated MODIS cloud productions, and also

demonstrates the scientific applications of the improved high-resolution retrievals.

Overall, it is a well-written, well organized paper, with an excellent balance of technical details and scientific applications. The documented methods can be applicable to many other instruments with similar features, e.g., VIIRS (I-band vs. M-band) and ABI. I think it is well suited for AMT and can be accepted largely as is, although a few minor comments/suggestions are listed below for the authors to consider to further improve the method/paper.

Minor comments:

Line: 19: MODIS team has an updated paper Platnick et al. 2017.Please update or add the reference

S. Platnick et al., "The MODIS Cloud Optical and Microphysical Products: Collection 6 Updates and Examples From Terra and Aqua," in IEEE Transactions on Geoscience and Remote Sensing, vol. 55, no. 1, pp. 502-525, Jan. 2017, doi: 10.1109/TGRS.2016.2610522.

The footnote about the numbers of Google scholar hits is very interesting! I guess the factor that Aqua-MODIS is part of A-train helps.

Page 4 around line 20: How well are the HRV band and other narrow bands spatially collocated, especially off the nadir region? For example, does a 3 x 3 km narrowband (e.g., 1.6 $\mu$m) pixel always contain 9 x 9 HRV pixels? If not, how are they collocated?

It is mentioned in Section 3.1 that, the cloud mask from NWC SAF is used. What is the spatial resolution for this cloud mask? Then it is also mentioned that a HRV-based loud mask is also used. How are the two cloud masks reconciled or combined?

Page 9 about LUT downscaling: There seems another way to do the downscaling, which is to assume the cloud effective radius remains invariant within the 3x3 km pixel. This seems to be easier than the slope-based Eq. (5). Can you comment on whether such method is feasible/practical or not and why?

On page 11, it is a little disappointing to see that the new method does not improve the CER retrievals. Nevertheless, some results of CER retrieval (e.g., a scatter plot or histogram) should be shown here. It is hard to picture the difference between SEVIRI and MODIS based on the description between line ∼20 to ∼30.

In addition to correlation, some more statistics should be added and discussed here, e.g., whether there is any systematic bias in CER? How about the extreme values?

One aspect missing in the discussion of COT and CER retrievals is about failed retrievals. As shown in Cho et al. 2015, MODIS retrievals frequently fail in broken cloud regions and/or at special angles (low sun, sunglint etc). Does the SEVIRI retrieval product also suffer from failed retrieval problems? If so, whether and how does the HRV alleviate the problems? Some discussions here would make the paper more interesting and useful.

Cho, H. M. et al. (2015), Frequency and causes of failed MODIS cloud property retrievals for liquid phase clouds over global oceans, Journal of Geophysical Research-Atmospheres, 120(9), 2015JD023161–n/a, doi:10.1002/2015JD023161.

---

## Referee Comment (RC2) · Anonymous Referee #2 · 21 Dec 2020

This paper describes a methodology to use the high-resolution visible (HRV) channel (1km resolution at nadir) on MSG SEVIRI to increase the spatial resolution of the standard cloud optical property retrievals (primarily optical thickness) from the 3km channels. The authors further show a brief comparison of optical thickness retrievals with those from Terra MODIS C6.1, and demonstrate the usefulness of the higher resolution retrievals in two applications, namely identification of convective initiation and calculations of surface solar irradiance. Both applications of SEVIRI cloud products are shown to improve with use of the HRV channel.

[Figure]

The focus of the paper isn't necessarily on the details of the approach itself, however, as these can be found in previous papers by the authors, nor on a thorough evaluation of the approach, which apparently can be found in a companion paper to this one. Rather, this paper simply seems geared towards demonstrating the potential of the HRV retrievals to improve applications of the SEVIRI cloud products. This is a useful goal. That said, the examples shown are more on an "operations" side rather than climate, so it's not clear how relevant such an approach will be beyond the lifetime of MSG given the approaching launch of MTG whose spatial resolution is an improvement even over the SEVIRI HRV. Moreover, while the authors claim applicability to the new advanced GEO imagers like GOES-R ABI and MTG FCI, it's not clear to me that a sophisticated approach like this one, essentially "pan-sharpening" coarser resolution narrowband VIS/NIR channels with a high resolution broader-band channel, is necessary since the high resolution VIS/NIR channels on those advanced imagers can likely be used directly to retrieve cloud optical properties. The authors do briefly mention facilitating a merged MSG/MTG climate data record in the last paragraph of the conclusion, but more in passing than anything else. In my opinion, the authors need to make a better case for the long-term usefulness of this approach, otherwise this may be viewed as just a novelty approach for MSG SEVIRI that might soon be obsolete. It's a well-written paper, though, and I think the case can be made, so I highly encourage the authors to give it some thought.

Comments

Page 2, line 18: Semantics here, but cloud products include more than just the optical properties listed.

Page 2, line 19: Please also add the more recent MODIS C6 paper (Platnick, S., Meyer, K. G., King, M. D., Wind, G., Amarasinghe, N., Marchant, B., et al. (2016). The MODIS Cloud Optical and Microphysical Products: Collection 6 Updates and Examples From Terra and Aqua. IEEE Transactions on Geoscience and Remote Sensing, 55(1), 502–525. http://doi.org/10.1109/TGRS.2016.2610522).

Page 2, line 20: In addition to biases and uncertainties, such effects can cause increased retrieval failures as well. See Cho, H.-M., Zhang, Z., Meyer, K., Lebsock, M., Platnick, S., Ackerman, A. S., et al. (2015). Frequency and causes of failed MODIS cloud property retrievals for liquid phase clouds over global oceans. Journal of Geophysical Research: Atmospheres, 120(9), 4132–4154. http://doi.org/10.1002/2015JD023161.

Page 2, lines 22-24: This mention of cloud droplet number concentration is unexpected here and not tied in to the rest of the paper. In fact, it's only mentioned here and somewhat offhand in the conclusion. It's thus a little irrelevant to this work.

Page 4, line 31: I assume the correction factors derived against MODIS account for spectral response differences?

Page 6, lines 21-22: Some more details on the phase algorithm would be nice here (e.g., how you get from the cloud types to thermodynamic phase), but it apparently doesn't play much role later in the paper so I'll leave it to the authors.

Page 6, lines 24-25: Since you're comparing retrievals to MODIS later on, the reader is left to assume that the single-scattering properties used here are consistent with the MODIS products. This of course is highly relevant to understanding the comparison. Please clarify.

Page 6, lines 28-29: Only radiometric uncertainty is accounted for? What about other error sources, such as ancillary data, forward models, etc.?

Page 7, line 32: I assume the second mention of CAMS in this sentence should actually refer to ECMWF, as in the previous sentence?

Fig. 5 and text on page 10, lines 21-29: Some sort of RGB would be useful to help interpret these optical thickness images. Also, do you mean nearest-neighbor sampling rather than interpolation? If interpolation, why is that necessary if you're only showing side-by-side image comparisons and scene statistics (histograms in Fig. 6) rather

than pixel-to-pixel comparisons? You might be smoothing the optical thickness field by interpolating, which may be a factor in the HRV retrievals seemingly being lower than MODIS (confirmed by the histograms in Fig. 6).

Page 10, lines 31-32: This is hard to tell from the color scheme in the histogram plot in Fig. 6, but it looks like the issue is only with too few optically thick clouds rather than too few optically thin.

Page 11, lines 3-8: The pixel sizes between MODIS and SEVIRI likely are different in this scene, though maybe not as different as you might think depending on where in the MODIS swath this region is – MODIS pixels grow to about 2x5km at the edge of swath. Also, you mention possible differences in algorithms, sensor calibration, and view geometry. Can you define what algorithm differences might cause retrieval differences? Sensor calibration differences are possible, though you mention earlier that SEVIRI observations have had correction factors applied that were derived against MODIS. Also, the angular differences may indeed be playing a role given the angular dependence of cloud reflection – what part of the scattering angle space are MODIS and SEVIRI sampling in this scene?

Page 11, lines 8-10: Could these differences in coverage be linked to differences in cloud mask results, with MODIS finding less clouds? A cloud mask plot would be illuminating. If not the cloud mask, then retrieval failures in MODIS are likely playing a role. You can verify this by looking at the Retrieval Failure Metric in the MOD06 files, which would also give you an estimate of what look-up table grid point optical thickness is closest to the out-of-solution space observation.

Page 11, line 18: I guess it isn't a surprise that effective radius retrievals do not improve, since, if I understand correctly, the only improvement would be due to the higher-resolution VIS/NIR reflectance changes aliasing into the effective radius retrievals due to the non-orthogonal solution space.

Page 12, lines 28-29: Why do you need to interpolate the standard retrievals to the HRV

grid for Fig. 8? This isn't a pixel-to-pixel comparison, so why not leave the retrievals at their native resolution for the statistics?

Page 12, lines 33-34: While the cloud optical thickness signature does appear earlier in the HRV retrievals, it's not clear in this discussion whether or not optical thickness is actually used in CI detection schemes. So it's hard to tell how relevant this improvement is.

Page 16, lines 3-5; lines 26-27: I don't think you would need this type of sophisticated approach for the GOES-R series, MTG FCI, or MODIS and VIIRS, since the highest resolution VIS/NIR channels can be used directly to retrieve cloud optical thickness, a different approach I think than that taken here.

Page 16, lines 27-30: This mention of climate applications makes the best case for the ongoing relevance of this sharpening approach, since I think it becomes obsolete with the new MTG FCI. The authors only showed operational applications that are undertaken in real time, rather than retrospective, so how useful this approach is in the future is unclear.
* * *

---

## Author Comment (AC1) · 23 Mar 2021

**Reply to Anonymous Referee #1**

We thank Anonymous Referee #1 very much for his/her encouraging review of our manuscript. We do however also realize based on her/him comments, that some aspects of our draft require further clarification, in particular the relation of this paper to the companion paper of Werner and Deneke (2020) (hereafter referred to as WD20) and other prior studies, as is evident from several of the comments raised by him/her.

We have adopted the following convention for our review: citations of the comments are given in italics, followed by our reply. Below each reply, a screen shot of the marked-up text modifications is given, generated with latexdiff. Deletions are shown in red/strike-through style, while insertions are underlined and shown in blue.

Please note that we have also considered several related comments by both reviewers on Sec.4.1 in combination, yielding a more substantial revision which can no longer can be directly associated with a single comment. An identical text listing the revisions including their rationale is included in the replies to both referees at the end.

**Reply to Specific Comments**

C1.1. Line: 19: MODIS team has an updated paper Platnick et al. 2017. Please update or add the reference.

Indeed, we unfortunately missed to reference this paper on the latest collection of the MODIS cloud products, as also noted by Referee #2 (see C2.2). We have added this as additional reference in the revised manuscript.

(Platnick et al., 2003). (Platnick et al., 2003, 2017).

**C1.2: The footnote about the numbers of Google scholar hits is very interesting! I guess the factor that Aqua-MODIS is part of A-train helps.**

We also found that a fun detail. From my perspective, MODIS in itself was an incredibly successful mission, maybe due to the fact that it made data access a lot easier than any previous satellite mission by having easily accessible products via the LAADS DAAC, but also due to the excellent work done by the various Science Teams towards providing high-quality/regularly improved products to end users. We have updated the query results in the footnote of the revised manuscript (NB: we plan to update these numbers once more for a final version of the manuscript).

1A search of Google Scholar TM, https://scholar.google.com for the term cloud combined with MODIS yields  $\frac{167.000}{196.000}$  hits, compared to  $\frac{58.300}{59.600}$  hits for the term AVHRR,  $\frac{10.800}{11.700}$  for SEVIRI, and  $\frac{9.930}{11.700}$  for VIIRS (Date of access: 3 August 2020March 2021)

C1.3: Page 4 around line 20: How well are the HRV band and other narrow bands spatially collocated, especially off the nadir region? For example, does a 3 x 3 km narrowband (e.g., 1.6  $\mu$ m) pixel always contain 9 x 9 HRV pixels? If not, how are they collocated?

Thank you for pointing out this important aspect, which has in fact been addressed in a previous paper by Deneke and Roebeling (2011). From the results given in that paper, the collocation of the HRV and the narrow-band channels agrees quite well, with systematic and random shifts of about 0.36km/0.1km(East-West vs. North South) +/- 0.1km, respectively (e.g. less than 10% of the optical resolution of a low-resolution pixel size). We consider this collocation sufficiently accurate for our purposes, in particular well above the target requirement of 0.6 km specified by EUMETSAT. While that paper attempted a correction for the remaining shift, this correction has not been done in the present study (it should however be rather simple to implement this correction in our processing). We have added the following text to the manuscript to clarify this point.

- 20 transform (see Deneke and Roebeling, 2010, for details). The difference of the MTFs of the HRV and the 0.6 μm channel is also plotted in the figure. It is used in our method to extract the high spatial frequency component contained in the HRV channel observations which is not resolved by the lower-resolution channels, as is described in detail in Sec. 3.5 of the paper. It should The combination of this high frequency component with the narrowband channels relies on a sufficiently accurate channel co-registration. For the solar channels, the co-registration accuracy is specified to be better than 0.6 km by EUMETSAT
- 25 (Schmetz et al., 2002). The results of Deneke and Roebeling (2010) suggest that the true accuracy is in fact significantly better, with a systematic and random magnitude of about 0.1 km and 0.3 km, respectively, and effects of misalignment are neglected here. It should also be noted that the optical resolution of the SEVIRI channels is lower than their sampling resolution by a factor of about 1.6, which can be seen by the significant attenuation of the frequency response well below the Nyquist limit.

**C1.4: It is mentioned in Section 3.1 that, the cloud mask from NWC SAF is used. What is the spatial resolution for this cloud mask? Then it is also mentioned that a HRV-based cloud mask is also used. How are the two cloud masks reconciled or combined?**

Re-reading the manuscript based on this comment, it became evident to us that too little information is been given to fully understand this aspect, and we have revised the text to more clearly describe the approach. The NWCSAF cloud mask has a  $3x3km^2$  resolution. The HRV-based mask is used subsequently to improve its resolution as a post-processing step, using a rule-based approach to combine the two outputs. The following 2 text sections have been change to better describe this approach:

**1. At the start of Section 3:**

neighbor (N.N.) interpolation is used for the cloud products and bilinear interpolation for the ancillary datasets. In addition, a high-resolution cloud masking algorithm is applied to the HRV channel reflectances, which has been introduced previously

30 by Bley and Deneke (2013), and is used to improve the standard-resolution NWCSAF cloud mask in particular with respect to the occurrence of small convective clouds as is described in Sec. 3.5. In the final step of the processing chain, these cloud

**2. At the beginning of Sect. 3.5:**

as coastal/shore. As the information content of the HRV channel for cloud masking is limited, a set of rules is used to combine

- 10 the standard- and high-resolution cloud masks, considering blocks of  $3 \times 3$  pixels, the HRV cloud mask and the NWCSAF cloud type. Semitransparent high clouds are frequently missed by the HRV cloud mask, and are thus labeled as cloudy based on the NWCSAF cloud type. The NWCSAF cloud mask does however frequently misclassify small clouds as cloud-free due to its coarser pixel resolution. Thus, pixels detected as cloudy by the HRV mask and as cloud-free by the NWCSAF mask are labeled as cloudy. The only exception to this rule are pixel blocks labeled as snow-covered land or sea-ice by the NWCSAF
- 15 cloud type algorithm, which are expected to be highly reflective and are thus left unchanged.

C1.5: On page 11, it is a little disappointing to see that the new method does not improve the CER retrievals. Nevertheless, some results of CER retrieval (e.g., a scatter plot or histogram) should be shown here. It is hard to picture the difference between SEVIRI and MODIS based on the description between line  $\sim$ 20 to  $\sim$ 30.

We agree that this is finding is somewhat disappointing, but it is also not unexpected (note however the different expectation expressed by referee #2, C2.14). The spectral response of the HRV channel only covers wavelengths within the conservative scattering regime. Hence, from a physical point of view, it cannot add a remote-sensing-based constraint on the effective radius. Thus, any improvement would have to come from a cloud-physics based constraint linking COT/CER, such as the adiabatic cloud assumption (see also C1.6). We would also like to stress that the aspect of CER quality is discussed more exensively in WD20, and it is shown there that a naive approach can even reduce the accuracy of CER retrievals. For concrete changes also prompted by this comment, see the description of the revisions to Sec.4.1 given below, which addresses the difference in CER in more detail.

C1.6: Page 9 about LUT downscaling: There seems another way to do the downscaling, which is to assume the cloud effective radius remains invariant within the 3x3 km pixel. This seems to be easier than the slope-based Eq. (5). Can you comment on whether such method is feasible/practical or not and why?

This assumption would force reflectances from a 3x3 pixel box to lie exactly on a CER contour in a classical Nakajima-King-style plot. Note that our current implementation does treat each HRV-resolution pixel independently / lower-resolution channels are interpolated to the HRV grid, which does not allow an exact implementation of this contraint, as the standard resolution channel radiances might thus vary across a 3x3 HRV pixel block. Nevertheless, the suggested approach has been investigated as one candidate approach in the companion paper by WD20. While we expected this (or the approach based on sub-adiabatic theory, see also reply to C1.5) to perform well/maybe even better than the method chosen here, the evaluation in WD20 showed otherwise. This finding is summarized by the following quote from WD20: "It is also an indication that assuming constant subpixel  $r_{eff}$  values within each LRES pixel is not sufficient." To explain better that this and other approaches have been tested in

WD20, and that the approach used is the one which has been found to perform best, we have revised the text as follows:

While it is simplest The simplest approach to implement the cloud retrievals based on is the assumption that  $\delta r_{16} = 0$ , i.e., that the high-frequency residual of the absorbing SWIR channel reflectance can be neglected, this assumption has been found. This assumption has however been observed to cause a degraded reduced accuracy of the retrieved effective radius

10 even effective radius in comparison to the standard resolution retrievals (see Werner and Deneke (2020) and the discussion in Sec. 4.1). Two other candidate approaches were considered in this study but found to be sub-optimal: the assumption of an adiabatic cloud, and that the effective radius remains constant within a 3 × 3 pixel neighbourhood.

**[Figure 4 about here.]**

Instead, the retrieval has been modified to determine approach determined to be most accurate by Werner and Deneke (2020)

15 is used here: it determines the high-frequency residual  $\delta r_{16} = 0$  based on the tangent of the  $\tau$ -contour at the location of the standard-resolution reflectances in the Nakajima-King diagram (referred to as lookup-table approach with slope adjustment in Werner and Deneke (2020)). Mathematically, this can be expressed as the slope of the  $\tau$ -contour at the point  $\mathcal{F}(\tilde{\tau}, \tilde{r}_e)$ , or

*C1.7:* In addition to correlation, some more statistics should be added and discussed here, e.g., whether there is any systematic bias in CER? How about the extreme values? See our description of the revisions to Sec.4.1 given below, which addresses this point.

C1.8: One aspect missing in the discussion of COT and CER retrievals is about failed retrievals. As shown in Cho et al. 2015, MODIS retrievals frequently fail in broken cloud regions and/or at special angles (low sun, sunglint etc). Does the SEVIRI retrieval product also suffer from failed retrieval problems? If so, whether and how does the HRV

**alleviate the problems? Some discussions here would make the paper more interesting and useful.**

See our description of the revisions to Sec.4.1 given below, which now also addresses this point.

**Revision of Sec.4.1:** Based on comments C1.5, C1.7, C1.8 by referee #1, as well as comments C2.3, C2.7, C2.10, C2.11 by referee #2, we have decided to substantially revise the presentation of Sec.4.1, with the following objectives:

- Add a description of the observing and sun geometries, including the true MODIS spatial resolution
- Discuss discrepancies in retrieval assumptions/conditions by MODIS/CPP, in particular including the width of the cloud drop size distribution and its relevance close to the cloud bow
- Mention the frequency of retrieval failures in both MODIS and CPP retrievals
- More detailed discussion of the accuracy of CER, which is known to be limited for such types of cloud fields, and point out the limitations of a comparison based on a single scene.
- Added an RGB image as 4th panel of the scene to Fig.5
- Revised Fig.6 to use separate panels/also show MODIS partially cloud retrievals
- Remove the erroneous interpretation of Fig.6 that SEVIRI retrieves too few optically thin clouds

The revised sub-section 4.1 is appended to this reply.

[revised manuscript text omitted]

---

## Author Comment (AC2) · 23 Mar 2021

**Reply to Anonymous Referee #2**

We also thank Anonymous Referee #2 very much for his/her constructive and critical comments, which hopefully helped to significantly improve the presentation of our paper. In the reply, we hope in particular to convince her/him of the long-term usefulness of this approach, as we do not believe that it will become entirely obsolete with the arrival of MTG.

We have adopted the following convention for our review: citations of the comments are given in italics, followed by our reply. Below each reply, a screen shot of the marked-up text modifications is given, generated with latexdiff. Deletions are shown in red/strike-through style, while insertions are underlined and shown in blue.

Please note that we have also considered several related comments by both reviewers on Sec.4.1 in combination, yielding a more substantial revision which can no longer can be directly associated with a single comment. An identical text listing the revisions including their rationale is included in the replies to both referees at the end.

Specific comments:

**C2.1: Page 2, line 18: Semantics here, but cloud products include more than just the optical properties listed.**

Yes, thanks for pointing this out. We do however wanted to refer here specifically to the properties obtained from Nakajima-King style retrievals. We have tried to clarify this aspect by the following text changes (page 2, line 17ff):

Currently, the Moderate Resolution Imaging Spectroradiometer (MODIS) flown on the polar-orbiting satellites Terra and Aqua are one of the most widely used satellite instruments for studying the role of clouds in the climate 1. Based on the A subset of the MODIS cloud products is based on solar reflectances and the bi-spectral method described by Nakajima and King (1990), cloud products (e.g., estimates of cloud phase, optical depth, effective radius, and cloud-water path) are provided at a
 spatial resolution of ~ 1 × 1 km2 (Platnick et al., 2003), (Platnick et al., 2003, 2017).

C2.2.: Page 2, line 19: Please also add the more recent MODIS C6 paper.

Indeed, we unfortunately missed this paper (see also C1.1). We have added this reference in the revised manuscript (this change is included in text changes shown for C2.1).

C2.3: Page 2, line 20: In addition to biases and uncertainties, such effects can cause increased retrieval failures as well.

Indeed, we missed to mention this important aspect. The text change to add this point is shown below. Note that this aspect is also now discussed in more detail in our revised version of Sec.4.1 as is described below.

Despite their wide use, it is well recognized well-recognized that sub-pixel variability and 3D radiative effects can cause retrieval failures (Cho et al., 2015), or introduce substantial biases and uncertainties in these products, which depend on various factors such as solar and viewing geometry (see e.g., Cahalan et al., 1994; Marshak et al., 2006; Zhang et al., 2012; Horváth

C2.4: Page 2, lines 22-24: This mention of cloud droplet number concentration is unexpected

**here and not tied in to the rest of the paper. In fact, it's only mentioned here and somewhat offhand in the conclusion. It's thus a little irrelevant to this work.**

Cloud droplet number (CDN) is retrieved in various studies based on satellite-retrieved COT and CER (e.g. Quaas et al., 2006). Based on the relation to COT/CER used in that study, it is easy to obtain estimated from the HRV-based COT/CER retrieval, and the accuracy has been discussed in WD20. As cloud-aerosol interactions are one of the scientific interests of the first author, we have chosen keep the discussion on CDN, but have attempted to better clarify its connection to the present paper (page 2, lines 22ff).

et al., 2014). While the retrieval of cloud droplet number concentration is of high scientific interest due to its relevance for
elucidating the climate impact of aerosol-cloud interactions, it is particularly challenging its estimation from bi-spectral solar
reflectances by means of cloud optical depth and effective radius (see e.g. Quaas et al., 2006) is particularly sensitive to such uncertainties (Grosvenor et al., 2018).

**C2.5:* Page 4, line 31: I assume the correction factors derived against MODIS account for spectral response differences?**

Instead of an answer, a quote from Meirink et al., 2013 is given here: "Specific attention is paid to correcting for differences in spectral response between instruments." We have revised the manuscript to clarify this aspect:

et al., 2004). Comparing collocated near-nadir reflectances from the SEVIRI and MODIS instruments and accounting for differences in spectral response between the instruments, Meirink et al. (2013) confirm the temporal stability of this calibration, but find relatively large systematic differences of up to 8% for collocated near-nadir reflectances from the SEVIRI and MODIS

5 instruments. Channel-specific correction factors to account for these differences have been derived and are applied by the

C2.6: Page 6, lines 21-22: Some more details on the phase algorithm would be nice here (e.g., how you get from the cloud types to thermodynamic phase), but it apparently doesn't play much role later in the paper so I'll leave it to the authors.

We have added a more detailed description of the cloud phase algorithm to Sec.3.2, which is shown here:

Retrievals are performed for assuming either a liquid or an ice cloud, based on a determination of the cloud thermodynamic phase by a modified version of the Pavolonis et al. (2005) algorithm. Several spectral tests are performed based on observed SEVIRI brightness temperatures at 6.2, 8.7, which is described in more detail in 10.8, 12.0 and 13.4  $\mu$ m, as well as clear and cloudy sky IR brightness temperature simulations with RTTOV. The algorithm initially yields 5 cloud types, which are further

5 condensed into a classification of liquid and ice phase. More details are given in Benas et al. (2017).

**C2.7:* Page 6, lines 24-25: Since you're comparing retrievals to MODIS later on, the reader is left to assume that the single-scattering properties used here are consistent with the MODIS products. This of course is highly relevant to understanding the comparison. Please clarify.**

This is an interesting point, however we do think this point is out of scope/cannot be covered comprehensively within the current paper (there is a whole sub-group of the CREW/ICWG initiative dedicated to this aspect). For water clouds, the single-scattering properties of cloud droplets should be rather consistent, but the assumed width of the droplet size distribution does introduce some differences (see Benas et al. 2019). However, this is only expected to matter for geometries close to the cloud bow. Assumptions about ice cloud properties are however a much less consistent, which will impact a direct comparison, but we are not considering ice clouds here for this reason.We have decided to carry out

CPP retrievals based on MODIS data in WD20 to avoid this point. Within the scope of the present paper, we have decided to make the following two changes:

- Add information on the assumed droplet size distribution/width used by the CPP retrieval in Sec.3.2, see below.
- Mention the width used by MODIS C6 retrievals, and its potential impact on the comparison as part of the revisions to Sec.4.1 described separately.

CPP employs precalculated lookup tables (LUTs) of TOA cloud-top reflectances in a Rayleigh atmosphere, which have been simulated by the Doubling-Adding KNMI (DAK) radiative transfer model (Stammes, 2001). Details For water clouds, the droplet size distribution is assumed to follow a two-parameter Gamma function with an effective variance of 0.15, while randomly oriented monodisperse imperfect hexagonal crystals are assumed for ice clouds. More details on the underlying

10 single-scattering properties of liquid and ice the cloud particles can be found in Benas et al. (2017). The measured reflectances

**C2.8: Page 6, lines 28-29: Only radiometric uncertainty is accounted for? What about other error sources, such as ancillary data, forward models, etc.?**

Yes, this is a limitation of the current implementation of the error estimates by the CPP algorithm. While a more comprehensive uncertainty estimate is obviously desirable, this is beyond the scope of the current study and left for future work.

**C2.9: Page 7, line 32: I assume the second mention of CAMS in this sentence should actually refer to ECMWF, as in the previous sentence?**

No, CAMS is indeed correctly named as source. We are using the ECMWF forecast for atmospheric profiles&surface temperature, but ECMWF does not provide any aerosol forecasts. Hence, aerosol properties are obtained from the CAMS forecast for near-real time application. Re-reading this section, we have found the following text to be misleading / incorrect, likely prompting above comment, and have thus changed it as follows:

transfer model. As input data streams, either the CAMS (Copernicus Atmospheric Monitoring Service) reanalysis (Inness et al., 2019) or the operational ECMWF forecast together with aerosol properties from the CAMS forecast can be used alternatively with 3-hourly resolution, the latter allowing for near-real time processing, while the former is only available with some time
delay. In addition, the NWCSAF software uses the OSTIA dataset as input for sea surface temperature (Donlon et al., 2012).

*C2.10:* Fig. 5 and text on page 10, lines 21-29: Some sort of RGB would be useful to help interpret these optical thickness images. Also, do you mean nearest-neighbor sampling rather than interpolation? If interpolation, why is that necessary if you're only showing side-by-side image comparisons and scene statistics (histograms in Fig. 6) rather than pixel-to-pixel comparisons? You might be smoothing the optical thickness field by interpolating, which may be a factor in the HRV retrievals seemingly being lower than MODIS (confirmed by the histograms in Fig. 6).

We believe that your term "nearest-neighbor sampling" actually is identical in meaning to what we call "nearest-neighbor interpolation". After reviewing e.g. image processing literature, we do think that our terminology is consistent with other use, thus have decided to not change the text. Nevertheless, NN interpolation does not smooth the field and thus can be ruled out as reason for differences in these histograms. We have also decided to add an RGB to Fig.5, see our revisions to Sec.4.1.

*C2.11:* Page 10, lines 31-32: This is hard to tell from the color scheme in the histogram plot in Fig. 6, but it looks like the issue is only with too few optically thick clouds rather than too few optically thin.

We have changed the visualization of the histogram to use 3 separate panels, see revisions to Sec.4.1. Indeed, after re-viewing the text, we agree that the "too few optically thin clouds" was an erroneous interpretation, and is interpretation has been removed as part of our revisions to Sec.4.1.

C2.12: Page 11, lines 3-8: The pixel sizes between MODIS and SEVIRI likely are different in this scene, though maybe not as different as you might think depending on where in the MODIS swath this region is – MODIS pixels grow to about 2x5km at the edge of swath. Also, you mention possible differences in algorithms, sensor calibration, and view geometry. Can you define what algorithm differences might cause retrieval differences? Sensor calibration differences are possible, though you mention earlier that SEVIRI observations have had correction factors applied that were derived against MODIS. Also, the angular differences may indeed be playing a role given the angular dependence of cloud reflection – what part of the scattering angle space are MODIS and SEVIRI sampling in this scene?

We did select this scene to be close to nadir-viewing (~  $3^{\circ}$  in this case at center) to mitigate the mentioned effect. This translates to 1007m and 1006m pixel size in across/along-track direction. MODIS is viewing the scene at ~ 154° scattering angle (180° is backward scattering), SEVIRI around ~150°, far enough away from the cloud bow at 141°. We have now revised Sec.4.1 to give more information on these details, see description of the revisions.

C2.13: Page 11, lines 8-10: Could these differences in coverage be linked to differences in cloud mask results, with MODIS finding less clouds? A cloud mask plot would be illuminating. If not the cloud mask, then retrieval failures in MODIS are likely playing a role. You can verify this by looking at the Retrieval Failure Metric in the MOD06 files, which would also give you an estimate of what look-up table grid point optical thickness is closest to the out-of-solution space observation.

Actually, a common pixel mask based on the requirement of COT>0.1 has been used to rule out such an effect, but this was unfortunately not mentioned in the discussion paper. This omission has now been corrected. MODIS does indeed "see" less clouds (83% cloud coverage vs. ~ 98% for both standard-and HRV resolution retrievals), likely due to the effect of the oblique satellite viewing angle of SEVIRI in combination with the larger pixel size.

We have now revised Sec.4.1 to give more information on these details, see description of the revisions. In particular, we have imposed common selection criteria to avoid any influence of cloud mask/retrieval failures on these results. A corresponding description has been added to the revised Sub-Section 4.1, to make it clearer that these reasons can be ruled out.

C2.14: Page 11, line 18: I guess it isn't a surprise that effective radius retrievals do not improve, since, if I understand correctly, the only improvement would be due to the higher resolution VIS/NIR reflectance changes aliasing into the effective radius retrievals due to the non-orthogonal solution space.

It is interesting to note that this comment is based on a rather different expectation than the comment C1.5 by referee #1. We have commented on this aspect in our revisions to Sec.4.1, which are shown separately.

C2.15: Page 12, lines 28-29: Why do you need to interpolate the standard retrievals to the HRV grid for Fig. 8? This isn't a pixel-to-pixel comparison, so why not leave the retrievals at their native resolution for the statistics?

The challenge arises due to the small size of the convective cell. Our intention is to get a better/fairer comparison both visually and statistically, as simple interpolation approaches such as bilinear or bicubic interpolation are readily available. Leaving the retrievals at native resolution, there is a much smaller number of pixels due to the small size of the developing convective cell (just 2-3 pixels up to

13:45Z). Note that we have also used infrared brightness temperature field interpolated to the HRV grid to define the cloud objects, because at standard-resolution, large gradients in this field complicate the separation of cloud objects from cloud-free background, yieldingsignificantly worse separation for the HRV retrieval results. We attach here the same plot just using the native resolution retrievals, which we hope supports our choice visually (alternatively, if there are any convincing arguments otherwise which we have missed, this could be used as replacement).

**C2.16:* Page 12, lines 33-34: While the cloud optical thickness signature does appear earlier in the *HRV* retrievals, it's not clear in this discussion whether or not optical thickness is actually used in *CI* detection schemes. So it's hard to tell how relevant this improvement is.**

While we are not aware how common the use of cloud properties is versus the use of radiances in operational satellite-based CI schemes, we do believe that the use of cloud properties has advantages for physical understanding, and has been discussed in the literature. To support this aspect, we have made the following revision to the manuscript:

channels, and are thus limited to the standard spatial resolution of SEVIRI (e.g. Mecikalski et al., 2010). Some previous studies already point out some-benefits arising from the use of the HRV channel (e.g. Carbajal Henken et al., 2011; Mecikalski et al., 2013a; Merk and Zinner, 2013). Its-While the first satellite-based CI detection schemes have been based on observed radiance fields (Mecikalski and Bedka, 2006), the use cloud properties instead of radiances seem promising, as it should

25 remove co-variability with environmental influences and viewing geometry, and aid a more physical interpretation of cloud growth (e.g. Senf and Deneke, 2017). It has thus been considered in several more recent scientific studies (e.g. Mecikalski et al., 2011, 2013). The combination of cloud products and higher spatial resolution offered by the HRV channel could thus be one way to improve the lead time for the detection of country initiation in surrent MSC based CL detection schemes. In particular, developing

the lead time for the detection of convective initiation in current MSG-based CI detection schemes. In particular, developing

C2.17: Page 16, lines 3-5; lines 26-27: I don't think you would need this type of sophisticated approach for the GOES-R series, MTG FCI, or MODIS and VIIRS, since the highest resolution VIS/NIR channels can be used directly to retrieve cloud optical thickness, a different approach I think than that taken here.

There are two important parts of our algorithm for carrying out the HRV-resolution cloud property retrievals in the current MSG-based scheme:

- The first part is the use of a linear relation for linking HRV, 0.6 and 0.8um reflectance variations to overcome the challenge posed by the large spectral width of the HRV channel (and consistency with 0.6um-based retrievals), which is indeed MSG-specific and will become obsolete with MTG. This part was already covered in-depth in Deneke and Roebeling (2010), so this is not really the new aspect covered by our paper and WD20.
- The use of multi-resolution satellite images for Nakajima-King retrievals, specifically with a VIS channel at higher spatial resolution than the SWIR channel, including its use for cloud masking (here based on the approach described previously by Bley and Deneke, 2013). This part is also applicable to current-generation sensors (MODIS: 0.6um channel at 0.25km, 1.6um at 0.5km, and 2.2um at 1km), GOES-R ABI (0.6um channel at 0.5km, 1.6um at 1km, and 2.2um at 2km) and will be applicable to MTG (0.6 and 0.8um channels at 500m, 1.6um and 2.2um at 1.0km). (VIIRS is rather an exception, which offers all required channels for such retrievals at 375m resolution. We erroneously included it in the list and have removed it in our revision). Thus, Nakajima-King-style retrievals can be carried out with the same or similar constraints used for handling small-scale variability in the SWIR channel in the retrieval at the spatial resolution of the highest-resolution VIS channel, and this aspect has to our knowledge not been covered before in the scientific literature. And one central message of our studies (both WD20 and this paper) is that care must be taken to not degrade the accuracy of the CER retrievals (see also reply to referee #1, C1.6, as well as the results of WD20).

To clarify this aspect, we have made the following revisions to the conclusion:

Similar approaches could also be adopted. The method described here and in Werner and Deneke (2020) can also be adapted to other meteorological multi-resolution imagers such as the Advanced Baseline Imager aboard the current generation of geostationary GOES satellites, or the the Flexible Combined Imager on the upcoming Meteosat Third Generation, and the polar-orbiting MODIS and VIIRS sensors, and MODIS instruments. This would allow to increase the spatial resolution of the retrieved cloud and radiation products up to that offered by the highest-spatial resolution channels. It has however to be 500 m for GOES-R and MTG, and to 250 m for MODIS. These instruments have in common that they feature a higher-spatial

- 5 resolution VIS channel, which allows to constrain cloud optical depth (e.g. the  $0.6 \mu m$  channel with 250 m and 500 m resolution for MODIS and GOES-R ABI, respectively), and lower-resolution SWIR channels (e.g. 1.6 or  $2.2 \mu m$ ) used as constraint for the cloud effective radius in bi-spectral retrievals. In fact, the constraint imposed due to the relatively broad spectral width of the HRV channel and formalized by Eq. 3 would no longer be required. Only the constraint on the small-scale variability in SWIR reflectance and expressed by Eq. 5 would be needed, thus allowing a simplified implementation. It should be noted,
- 10 however, that based on the findings presented here and in Werner and Deneke (2020), a too simplistic constraint as e.g. realized by simple interpolation of the absorbing-channel reflectance will lead to a reduced accuracy for cloud effective radius, which might in fact be worse than that of a retrieval done at the lowest common spatial resolution. It also has to be cautioned that a higher spatial resolution does not necessarily imply a higher product accuracy. Specifically, the findings of Zinner and Mayer

C2.18: Page 16, lines 27-30: This mention of climate applications makes the best case for the ongoing relevance of this sharpening approach, since I think it becomes obsolete with the new MTG FCI. The authors only showed operational applications that are undertaken in real time, rather than retrospective, so how useful this approach is in the future is unclear.

See our response to C2.17, we do not think the approach described here will become obsolete. We are aware that the reference to potential climate applications is somewhat speculative, and that we do not provide any examples of climate-relevant applications. Nevertheless, we hope this statement might

raise interest in using our method for climate applications, and might lead to future collaboration on such an application. Hence, we have decided to leave this statement here in unmodified form.

**Revision of Sec.4.1:** Based on comments C1.5, C1.7, C1.8 by referee #1, as well as comments C2.3, C2.7, C2.10, C2.11 by referee #2, we have decided to substantially revise the presentation of Sec.4.1, with the following objectives:

- Add a description of the observing and sun geometries, including the true MODIS spatial resolution
- Discuss discrepancies in retrieval assumptions/conditions by MODIS/CPP, in particular including the width of the cloud drop size distribution and its relevance close to the cloud bow
- Mention the frequency of retrieval failures in both MODIS and CPP retrievals
- More detailed discussion of the accuracy of CER, which is known to be limited for such types of cloud fields, and point out the limitations of a comparison based on a single scene.
- Added an RGB image as 4th panel of the scene to Fig.5
- Revised Fig.6 to use separate panels/also show MODIS partially cloud retrievals
- Remove the erroneous interpretation of Fig.6 that SEVIRI retrieves too few optically thin clouds

The revised sub-section 4.1 is appended to this reply.

[revised manuscript text omitted]

---

## Author Response (AR2)

**Reply to Anonymous Referee #2 and Editor**

We thank both Anonymous Referee #2 and the editor Dr. Jethva very much for their time and efforts to review our revised manuscript.

Regarding the comment by the editor:
> When going through the revised manuscript, it was noticed in Figure 5 that you might
> have labeled the sub-plot incorrectly. The improved METEOSAT SEVIRI retrievals are
> shown in (c), whereas the standard-resolution SEVIRI retrievals are shown in (d). A visual
> inspection of these figures tells that the standard-resolution 3x3 km-square retrievals are
> shown in (c) and proposed cloud retrievals are displayed in (d). Please double-check and
> let me know if my observation is correct or not.
Yes, we indeed refer here to the wrong figure panels, thanks for pointing out this mistake. We have decided to rectify this problem by swapping the panels (c) and (d) in the figure, so now the cited version of the figure caption is indeed correct.